# SKILL-BASED REINFORCEMENT LEARNING WITH INTRINSIC REWARD MATCHING

## ABSTRACT

While unsupervised skill discovery has shown promise in autonomously acquiring behavioral primitives, there is still a large methodological disconnect between task-agnostic skill pretraining and downstream, task-aware finetuning. We present Intrinsic Reward Matching (IRM), which unifies these two phases of learning via the *skill discriminator*, a pretraining model component often discarded during finetuning. Conventional approaches finetune pretrained agents directly at the policy level, often relying on expensive environment rollouts to empirically determine the optimal skill. However, often the most concise yet complete description of a task is the reward function itself, and skill learning methods learn an *intrinsic* reward function via the discriminator that corresponds to the skill policy. We propose to leverage the skill discriminator to *match* the intrinsic and downstream task rewards and determine the optimal skill for an unseen task without environment samples, consequently finetuning with greater sample-efficiency. Furthermore, we generalize IRM to sequence skills and solve more complex, long-horizon tasks. We demonstrate that IRM enables us to utilize pretrained skills far more effectively than previous skill selection methods on the Unsupervised Reinforcement Learning Benchmark and on challenging tabletop manipulation tasks.

## 1 INTRODUCTION

Generalist agents must possess the ability to execute a diverse set of behaviors and flexibly adapt them to complete novel tasks. Although deep reinforcement learning has proven to be a potent tool for solving complex control and reasoning tasks such as in-hand manipulation (OpenAI et al., 2019) and the game of Go (Silver et al., 2016), specialist deep RL agents learn each new task from scratch, possibly collecting new data and learning to a new objective with no prior knowledge. This presents a massive roadblock in the way of integration of RL in many real-time applications such as robotic control where collecting data and resetting robot experiments is prohibitively costly (Kalashnikov et al., 2018).

Recent progress in scaling multitask reinforcement learning (Reed et al., 2022; Kalashnikov et al., 2021) has revealed the potential of multitask agents to encode vast skill repertoires, rivaling the performance of specialist agents and even generalizing to out-of-distribution tasks. Moreover, skill-based unsupervised RL (Laskin et al., 2022; Liu & Abbeel, 2021; Sharma et al., 2020) shows promise of acquiring similarly useful behaviors but without the expensive per-task supervision required for conventional multitask RL. Recent skill-based RL results suggest that unsupervised RL can distill diverse behaviors into distinguishable skill policies; however, such approaches lack a principled framework for connecting unsupervised pretraining and downstream finetuning. The current state-of-the-art leverages inefficient skill search methods at the policy level such as performing a sampling-based optimization or sweeping a coarse discretization of the skill space (Laskin et al., 2021). However, such methods still exhibit key limitations, namely they (1) rely on expensive environment trials to evaluate which skill is optimal and (2) are likely to select suboptimal behaviors as the continuous skill space grows due to the curse of dimensionality.

In this work, we present Intrinsic Reward Matching (IRM), a scalable algorithmic methodology for unifying unsupervised skill pretraining and downstream task finetuning by leveraging the learned intrinsic reward function parameterized by the skill discriminator. Centrally, we introduce a novel approach to leveraging the intrinsic reward model as a multitask reward function that, via

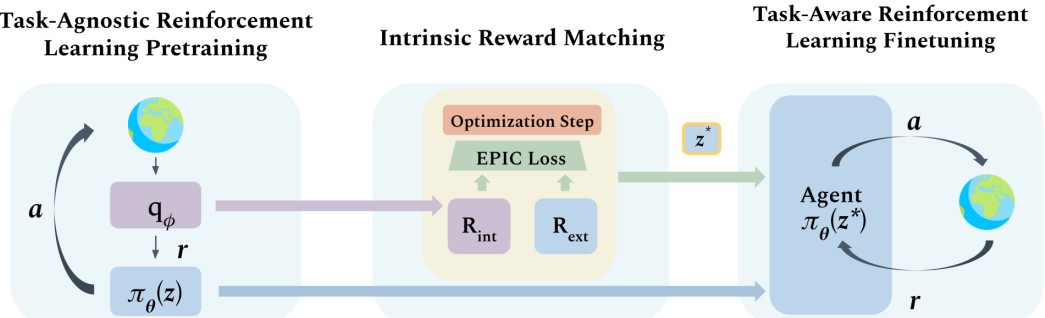

Figure 1: Intrinsic Reward Matching (IRM) Framework. IRM takes place in three stages: (1) Task-agnostic RL pretraining learns skill primitives in conjunction with a skill discriminator. (2) With no environment interaction, IRM minimizes the EPIC Loss between the intrinsic reward parameterized by the discriminator and the extrinsic reward with respect to the skill vector $z$. (3) The skill policy conditioned on the optimal $z^*$ finetunes to task reward to solve the downstream task.

interaction-free task inference, enables us to select the most optimal pretrained policy for the extrinsic task reward. During pretraining, unsupervised skill discovery methods learn a discriminator-parameterized, family of reward functions that correspond to a family of policies, or skills, through a shared latent code. Instead of discarding the discriminator during finetuning as is done in prior work, we observe that the discriminator is an effective task specifier for its corresponding policy that can be *matched* with the extrinsic reward, allowing us to perform skill selection while bypassing brute force environment trials. Our approach views the extrinsic reward as a distribution with measurable proximity to a pretrained multitask reward distribution and formulates an optimization with respect to skills over a reward distance metric called EPIC (Gleave et al., 2020).

**Contributions** The key contributions of this paper are summarized as follows: (1) We describe a unifying discriminator reward matching framework and introduce a practical algorithm for selecting skills without relying on environment samples (Section 3). (2) We demonstrate that our method is competitive with previous finetuning approaches on the Unsupervised Reinforcement Learning Benchmark (URLB), a suite of 12 continuous control tasks (Section 4.1). (3) We evaluate our approach on more challenging tabletop manipulation environments which underscore the limitations of previous approaches and show that our method finetunes more efficiently (Section 4.2). (4) We generalize our method to sequence pretrained skills and solve long-horizon manipulation tasks (Section 4.3) as well as ablate key algorithmic components. (5) We provide analysis and visualizations that yield insight into how skills are selected and further justify the generality of our method (Section 5).

## 2 BACKGROUND

### 2.1 UNSUPERVISED SKILL PRETRAINING

The skill learning literature has long sought to design agents that autonomously acquire structured behaviors in new environments (Thrun & Schwartz, 1994; Sutton et al., 1999; Pickett & Barto, 2002). Recent work in competence-based unsupervised RL proposes generic objectives encouraging the discovery of skills representing diverse and useful behaviors (Eysenbach et al., 2019; Sharma et al., 2020; Laskin et al., 2022). A skill is defined as a latent code vector $z \in \mathcal{Z}$ that indexes the conditional policy $\pi(a|s, z)$. In order to learn such a policy, this class of skill pretraining algorithms maximizes the mutual information between sampled skills and their resulting trajectories $\tau$ (Gregor et al., 2016a; Eysenbach et al., 2018; Sharma et al., 2019) :

$$I(\tau; z) = \mathcal{H}(z) - \mathcal{H}(z|\tau) = \mathcal{H}(\tau) - \mathcal{H}(\tau|z) \tag{1}$$

Since the mutual information $I(s; z)$ is intractable to calculate in practice, competence-based methods instead maximize a variational lower bound proposed in (Barber & Agakov, 2003) which is

parameterized by a learned neural network function $q_\phi(\tau, z)$ called a skill discriminator. This discriminator, along with other terms independent of $z$, parameterizes an intrinsic reward that the skill policy $\pi(\cdot|s, z)$ maximizes during pretraining. Given an unseen task specification, the agent needs to infer which skill will finetune to solve the task with the fewest samples. For more detailed explanations of the mutual information decompositions of various skill discovery algorithms, refer to Appendix A.2.

**Pretrained Multitask Reward Functions:** We observe that the intrinsic reward function learned during skill pretraining can be viewed as a multitask reward function, where the continuous skill code $z$ determines the task. In other words, we have some function:

$$\mathcal{R}^{int}(\tau, z) := \text{VLB}(\tau, z) \tag{2}$$

where $\text{VLB} \leq I(\tau, z)$ is the variational lower bound proposed in (Barber & Agakov, 2003) ($\tau$ is a trajectory representation such as $(s, s')$). Since skill discovery algorithms aim to maximize $I(\tau, z)$, we can view its parameterized lower bound VLB as a multitask reward function: scoring transitions based on their alignment with a skill code (Laskin et al., 2022).

## 2.2 EQUIVALENT-POLICY INVARIANT COMPARISON

We can formalize a general notion of reward function similarity by equivalent-policy invariant comparison (EPIC) as established in (Gleave et al., 2020). EPIC defines a distance metric between two reward functions such that similar reward functions induce similar optimal policies. We consider the case of action-independent reward:

$$D_{\text{EPIC}}(R_A, R_B) = \mathbb{E}_{s_P, s'_P \sim D_P, S_C, S'_C \sim D_C}[D_\rho(C(R_A)(s_P, s'_P, S_C, S'_C), C(R_B)(s_P, s'_P, S_C, S'_C))]. \tag{3}$$

where $D_\rho(X, Y) = \sqrt{\frac{1 - \rho(X, Y)}{2}}$ is the Pearson distance between two random variables $X$ and $Y$, $s_P, s'_P$ are samples from the Pearson distribution $D_P$, and $S_C, S'_C$ are batches sampled from the Canonical distribution $D_C$. We compute the Pearson distance over Pearson samples $s_P, s'_P$, with additional canonicalization with batches $S_c, S'_c$ to ensure invariance over constant shifts and scaling. The canonicalized reward function is defined as:

$$C(R)(s_P, s'_P, S_C, S'_C) = R(s_P, s'_P) + \mathbb{E}[\gamma R(s'_P, S'_C) - R(s_P, S'_C) - \gamma R(S_C, S'_C)] \tag{4}$$

where $R : S \times S \to \mathbb{R}$ is a reward function. The expectation is taken over the Canonical distribution $D_C$; for simplicity, we sample these batches $S_C, S'_C \sim D_C$ ahead of time. The canonicalization ensures *invariance to reward shaping* such that rewards that have different shaping but induce similar optimal policies are close in distance. In practice, the final term can be omitted as the Pearson correlation is *invariant to constant shifts and scaling*.

# 3 INTRINSIC REWARD MATCHING

## 3.1 TASK INFERENCE VIA INTRINSIC REWARD MATCHING

A multitask reward function that can supervise the learning of diverse behaviors is useful in its own right. However, in the case of skill-based RL, we have additionally learned a corresponding $\pi(a|s, z)$. Therefore, for any "task" that can be specified by our intrinsic reward function, we already have an optimal policy, so long as we condition on the corresponding skill. If we have learned a sufficiently diverse library of skills, we might expect that some of our skills share behavioral similarity to the optimal policy for the downstream task. It thus also holds that the corresponding intrinsic reward for that skill is a semantically similar task specification to the downstream task.

Given this interpretation of intrinsic reward, we posit that the task of identifying which our pretrained skills to apply to a downstream task can be reframed as inferring which task in our multitask reward function is most similar to the downstream task. Moreover, we should hope to find the skill code $z$ that produces the reward function most semantically aligned with the downstream task reward.

With this formalism, we can formulate the task inference problem as performing the following optimization:

$$z^* = \arg\min_z D_{\text{EPIC}}(R^{int}(\tau, z), R^{ext}(\tau)) \tag{5}$$

in order to find $z^*$ most aligned with the task reward. Moreover, Equation 5 performs a minimization of a novel loss we name the *EPIC loss* with respect to the skill parameter $z$. By EPIC's equivalence class invariance, we know that if the EPIC loss is small for some $z^*$, and $\pi(a|s, z^*)$ is near optimal for $R^{int}(\tau, z^*)$, then $\pi(a|s, z^*)$ approaches the optimal policy for the task as specified by $R^{ext}$. Notably, we *require access to the task reward function $R_{ext}$ to compute the EPIC loss.* Leveraging a known task reward function is a divergence from conventional skill selection methods.

**Computing $R^{int}$ during reward matching** During pretraining, for some methods such as (Laskin et al., 2022; Sharma et al., 2020), we require negative samples in order to compute the variational objective in Equation 2 and avoid a degenerate optimization where all embedded trajectories have high similarity with all skills. However, during selection when skills are fixed, the negative sampling component amounts to a reward offset which does not impact the task semantics. Furthermore, since we may not in general have access to a large amount of negative samples on a given downstream task, we choose to simplify the objective to the following:

$$\mathcal{R}^{int}(\tau, z) := \text{VLB}(\tau, z) \equiv q_\phi(\tau, z) \tag{6}$$

where $q_\phi$ is the skill discriminator. This parameterization of the intrinsic reward preserves the alignment semantics of VLB without the normalization by negative samples. For more details regarding the discriminator parameterization of the intrinsic reward for (Laskin et al., 2022; Sharma et al., 2020) refer to Appendix A.3 and Appendix A.4.

---

**Algorithm 1:** Intrinsic Reward Matching (IRM)

**Require:** Downstream task $\mathcal{T}$, $D_S$, $P_S$
**Require:** Pretrained policy $\pi_\theta(a|s, z)$, intrinsic reward $r_{int}(s, s', z)$, and extrinsic reward $r_{ext}(s, s')$ for $\mathcal{T}$.
**Require:** Optimization $N_{OP} = 5000$ steps and finetune $N_{FT} = 100K$ steps.
/* Skill Selection of $z^*$ via EPIC Loss                                   */
1 **for** $N_{OP}$ steps **do**
2      Sample a batch of Pearson samples $S_P, S_P' \sim D_P, D_P$.
3      Sample Canonical samples $S_C, S_C' \sim D_C, D_C$.
4      **for** $s_i, s_i'$ in $S_P, S_P'$ **do**
5          Calculate EPIC Loss as $D_{\text{EPIC}}(r_{int}(s_i, s_i', z), r_{ext}(s_i, s_i')) = D_\rho(C_{D_S}(R_A)(s_i, s_i', S_C, S_C'), C_{D_S}(R_B)(s_i, s_i', S_C, S_C'))$ as in Equation 3
6      **end for**
7      Take optimization step on batch with respect to $z$ (gradient descent, CEM step, etc.) as in Equation 5.
8 **end for**
9 Evaluate zero-shot performance and finetune RL agent for $N_{FT}$ steps with $z^*$ on downstream task $\mathcal{T}$

---

## 3.2 EPIC SAMPLE-BASED APPROXIMATION

We make a number of sample-based approximations of various unknown quantities in order to concretize the continuous optimization Equation 5 as a tractable loss minimization problem.

**Canonical State Distribution Approximation:** In order to canonicalize our reward functions, we estimate the expectation over the state and next state distributions with a sample-based average over 1024 samples. These distributions can be entirely arbitrary, though using heavily out-of-distribution samples with respect to pretraining can weaken the accuracy of the approximation. We choose to instantiate a uniform distribution bounded by known workspace constraints for both of these distributions.

**Sampling Distribution for Pearson Correlation:** We find that generating samples uniformly roughly within the environment workspace bounds, just as with the reward canonicalization, often leads to strong approximations. Furthermore, as both sample generation and relatively inexpensive function evaluation are independent of the online-finetuning phase, we can perform the full skill optimization as a self-contained preprocess to downstream policy adaptation without any environment samples. Rough knowledge of workspace bounds represents some amount of prior environment knowledge. We leave more general options such as sampling from a learned generative model over

trajectories encountered during pretraining or sampling from saved pretraining data to future work. We ablate various sampling distribution choices in Table 6 and present the full algorithm in detail in Algorithm 1.

### 3.3 GENERALIZATION TO SKILL SEQUENCING

Many realistic downstream tasks derive additional complexity from temporally extended planning horizons. In contrast to hierarchical reinforcement learning (HRL) approaches, which aim to stitch together pretrained skills at the policy level with a higher-level manager policy, we can extend the task matching framework of IRM to efficiently solve the problem of skill sequencing, entirely doing away with the manager policy. Consider the long-horizon setting where we have a sequence of reward functions to optimize over some task horizon $H$. Central to the finetuning problem is determining over what time intervals should potentially different pretrained skills be selected. In this work we predetermine a fixed skill horizon $\lfloor H/N \rfloor$ where $N$ is the number of rewards. This skill horizon could in principle be specified as a parameter and learned from the task reward signal.

Next, in order to perform skill selection over each time interval, we perform the IRM algorithm in parallel for each reward. We note the key assumption that IRM requires access to the reward functions for each of the subtasks. For example, for a sequential goal reaching task, we divide the episode into $N$ segments for each of the $N$ goals and corresponding goal-reaching rewards. We then perform the IRM skill selection algorithm for each reward to select the optimal skill over each interval. After selecting the skills, we freeze our selections and finetune the skill policies jointly.

## 4 EXPERIMENTS

In this section we aim to experimentally evaluate whether IRM improves the adaptation sample-efficiency of skill finetuning on a downstream reinforcement learning task as compared to baselines. For pretraining skills, we experiment with both the CIC (Laskin et al., 2022) and DADS (Sharma et al., 2020) algorithms. We consider *IRM Random* a version of IRM that randomly samples skills and picks the one with the lowest EPIC loss, *IRM CEM* which selects elites as those skills with the lowest EPIC loss, and *IRM Gradient Descent* which minimizes the EPIC loss using the Adam optimizer and uses backpropagation through the discriminator to regress the optimal skill.

**Environments** We evaluate IRM on URLB (Laskin et al., 2021), which consists of twelve downstream tasks in three challenging continuous control domains in the DMControl suite: Walker, Quadruped, and Jaco. We also design a reaching and a tabletop pushing environment in the OpenAI Gym Fetch environment (Brockman et al., 2016) with further details in Appendix A.5.

**Baselines** We benchmark many conventional finetuning approaches after a single skill pretraining phase of Contrastive Intrinsic Control (CIC) (Laskin et al., 2022). The *Grid Search (GS)* baseline coarsely sweeps each of 10 skills evenly from the all 0's skill vector to the all 1's skill vector and finetunes the skill which achieves the best evaluation reward over an

**Fetch Environment**

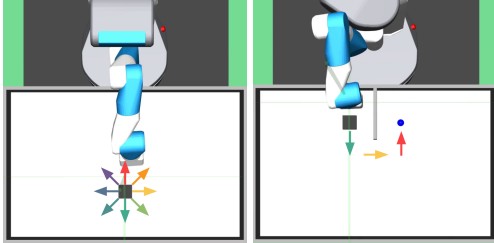

Figure 2: In our Fetch Push environment, we discover skills that move the block in different directions. Downstream tasks may involve simple goals or more distant goals that require composition of multiple skills across an extended time horizon and around obstacles.

episode. *Env Rollout* randomly samples 10 skills to evaluate with a rollout and *Env Rollout CEM* uses the episode reward as the metric by which to select elites. *Random Skill* selects a skill at random. *Relabel* relabels saved skill rollouts obtained during pretraining with the task reward function, and selects the skill that achieved the highest reward. All baselines use the TD3 (Fujimoto et al., 2018) RL algorithm.

**Evaluation** We follow an identical evaluation to the 2M pre-training setup in URLB. First, we pretrain each RL agent with the intrinsic rewards for 2M steps. Then, we finetune each agent to the downstream task with extrinsic rewards for 100k steps. Since our primary contribution involves

| Task | IRM CEM | IRM GD | IRM Rand | Env Roll. | Env CEM | GS | Relabel | Rand |
|---|---|---|---|---|---|---|---|---|
| Jaco Top Left | $0.000 \pm 0.00$ | $0.000 \pm 0.00$ | $0.000 \pm 0.00$ | $0.186 \pm 0.11$ | $0.770 \pm 0.28$ | $\mathbf{1.84} \pm \mathbf{0.00}$ | $0.000 \pm 0.00$ | $0.000 \pm 0.00$ |
| Jaco Top Right | $0.0860 \pm 0.040$ | $0.640 \pm 0.24$ | $0.120 \pm 0.097$ | $7.34 \pm 3.4$ | $9.82 \pm 5.3$ | $\mathbf{16.1} \pm \mathbf{0.00}$ | $0.000 \pm 0.00$ | $3.50 \pm 2.5$ |
| Jaco Bot. Left | $0.0520 \pm 0.030$ | $0.000 \pm 0.00$ | $0.000 \pm 0.00$ | $0.175 \pm 0.16$ | $\mathbf{0.408} \pm \mathbf{0.22}$ | $0.102 \pm 0.00$ | $0.000 \pm 0.00$ | $0.000 \pm 0.00$ |
| Jaco Bot. Right | $2.48 \pm 2.2$ | $0.000 \pm 0.00$ | $0.360 \pm 0.31$ | $0.086 \pm 0.073$ | $\mathbf{9.07} \pm \mathbf{3.3}$ | $0.191 \pm 0.00$ | $0.000 \pm 0.00$ | $0.00100 \pm 0.0010$ |
| **Walker Stand** | $\mathbf{19.9} \pm \mathbf{9.3}$ | $9.75 \pm 1.4$ | $12.5 \pm 3.0$ | $18.9 \pm 3.7$ | $\mathbf{22.4} \pm \mathbf{4.3}$ | $13.9 \pm 4.4$ | $3.00 \pm 0.18$ | $20.8 \pm 7.6$ |
| **Walker Walk** | $5.86 \pm 0.34$ | $7.48 \pm 0.55$ | $\mathbf{15.5} \pm \mathbf{5.5}$ | $14.9 \pm 2.9$ | $13.3 \pm 3.2$ | $9.40 \pm 2.8$ | $4.99 \pm 1.3$ | $\mathbf{15.6} \pm \mathbf{4.9}$ |
| Walker Run | $6.82 \pm 0.66$ | $7.17 \pm 0.28$ | $8.10 \pm 0.97$ | $7.92 \pm 0.69$ | $5.87 \pm 1.2$ | $6.56 \pm 1.2$ | $2.67 \pm 0.25$ | $\mathbf{8.81} \pm \mathbf{1.2}$ |
| Walker Flip | $20.6 \pm 1.2$ | $14.8 \pm 1.1$ | $17.3 \pm 2.3$ | $\mathbf{23.8} \pm \mathbf{1.9}$ | $17.3 \pm 2.8$ | $21.8 \pm 0.00$ | $3.29 \pm 0.00$ | $14.4 \pm 1.8$ |
| **Quadr. Stand** | $51.5 \pm 12$ | $40.3 \pm 11$ | $40.2 \pm 13$ | $40.6 \pm 9.7$ | $47.5 \pm 8.7$ | $37.4 \pm 12$ | $\mathbf{56.1} \pm \mathbf{11}$ | $44.6 \pm 13$ |
| **Quadr. Run** | $23.9 \pm 5.5$ | $\mathbf{24.4} \pm \mathbf{4.8}$ | $20.2 \pm 6.5$ | $20.6 \pm 4.4$ | $24.2 \pm 4.1$ | $17.3 \pm 5.5$ | $\mathbf{24.9} \pm \mathbf{6.5}$ | $21.7 \pm 6.2$ |
| **Quadr. Jump** | $38.1 \pm 8.5$ | $\mathbf{41.1} \pm \mathbf{9.4}$ | $35.9 \pm 11$ | $30.5 \pm 7.0$ | $36.8 \pm 6.3$ | $29.7 \pm 9.8$ | $35.5 \pm 9.0$ | $33.3 \pm 9.5$ |
| Quadr. Walk | $17.5 \pm 6.2$ | $11.5 \pm 3.7$ | $17.1 \pm 6.2$ | $19.3 \pm 2.7$ | $25.5 \pm 4.0$ | $9.21 \pm 2.0$ | $\mathbf{31.3} \pm \mathbf{8.2}$ | $16.4 \pm 5.8$ |
| Fetch Reach | $95.9 \pm 1.0$ | $87.5 \pm 0.20$ | $92.5 \pm 1.1$ | $85.0 \pm 6.2$ | $87.8 \pm 1.9$ | $\mathbf{97.3} \pm \mathbf{0.00}$ | $43.9 \pm 0.00$ | $16.7 \pm 19$ |
| **Fetch Push** | $\mathbf{80.2} \pm \mathbf{2.5}$ | $73.1 \pm 0.48$ | $77.6 \pm 2.7$ | $74.3 \pm 0.92$ | $75.4 \pm 2.6$ | $72.1 \pm 0.00$ | $23.6 \pm 0.023$ | $51.5 \pm 12.5$ |

Table 1: IRM with various optimization methods compared to environment rollout-based skill selection, reward relabelling of pretraining data, and random skill selection. IRM based methods rival or exceed skill selection baselines that are reliant on expensive environment trials.

skill selection, we especially focus on zero-shot episode rewards: rewards achieved by a selected skill policy but without any RL updates on the task reward. We report results averaged over 5 seeds with standard error bars.

### 4.1 Unsupervised Reinforcement Learning Benchmark

In Table 1, we display the zero-shot performance of IRM-based methods compared to interaction-based methods over all 12 URLB tasks. On most of the Walker and Quadruped tasks IRM is either comparable to or outperforms the interaction baselines. Reward relabelling fails to consistently select optimal skills across the benchmark, likely because its options are limited to the finite set of skills sampled during pretraining. IRM by contrast leverages continuous optimization in the skill space to find the best skill for the task. An important insight is that IRM uses the environment interactions to immediately begin finetuning the selected skill policy instead of spending significant amounts of samples on skill selection. This allows IRM-based methods to obtain greater sample-efficiency than rollout-based methods, even when both initial skill selections obtain similar performance as demonstrated in Figure 7. Unsurprisingly, methods like IRM GD and IRM CEM tend to perform better than IRM Random which does not have the luxury of iterative refinement on a smooth EPIC loss manifold as shown in Figure 5. We find that neither our method nor the baselines are well-suited for skill selection on the Jaco tasks. This is likely because these tasks are *very* sparsely rewarded, making it unlikely that many samples, either randomly generated as in IRM or rolled out, will consistently result in high rewards. We provide analysis demonstrating the relationship between task reward sparsity and the smoothness of the EPIC loss manifold in Appendix A.12.3.

### 4.2 Tabletop Manipulation

**Reach Target** We evaluate IRM on the Reach Target task, where the Fetch robot is rewarded for reaching a target position. IRM outperforms or closely matches environment-rollout methods while requiring no environment samples to perform skill selection. As shown in Table 1, the random skill policy performs particularly poorly and with very high variance relative to the IRM and environment-rollout based methods. Moreover, appropriate skill selection is required for strong zero-shot performance as certain skills obtain much higher rewards than others. Figure 3 shows the finetuning performance of the methods on the downstream task reward. IRM-based methods are more sample efficient in reaching the optimal performance than environment-rollout-based methods due to improved skill selection.

**Push Block to Goal** Next, we evaluate IRM on a more complex manipulation task involving pushing a block to a goal position. We report the zero-shot IRM skill selection performance in Table 1 and finetuning performance in Figure 3. This more complex task similarly benefits from bootstrapping the appropriate pretrained skill policy as evidenced by the performance gap of the selection based methods over random skill selection. We remark that even for more complex manipulation tasks, IRM is robust in consistently guiding optimal skill selection without requiring any interaction with the environment. Although Env Rollout CEM is one of the stronger baselines in terms of zero-shot reward, it exceeds the computational budget of 100k interactions entirely on

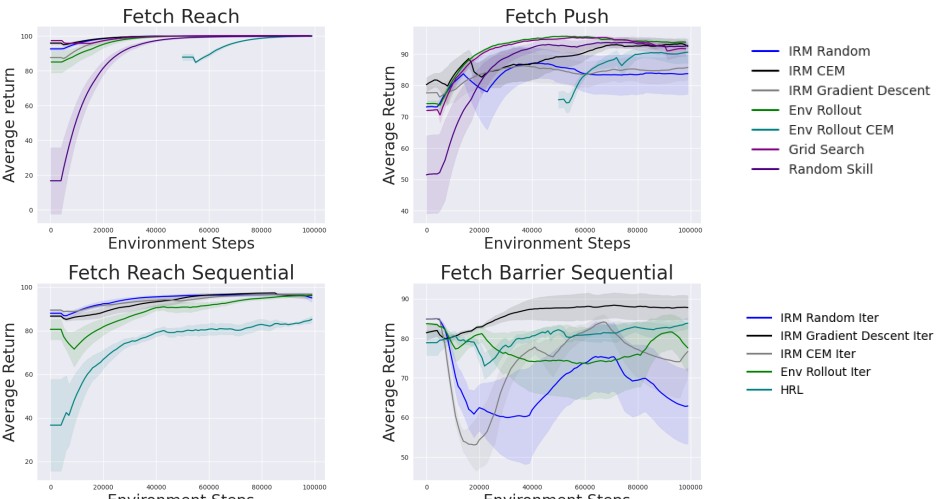

Figure 3: The performance gap between the IRM skill selection methods and random skill selection evidences the sample efficiency gains to be had from bootstrapping a pretrained policy with task-level semantics similar to the task reward. IRM-based methods select optimal skills with no environment interaction and consequently finetune efficiently. **Top:** Fetch Reach and Block Push tasks. **Bottom:** Long-horizon Fetch Reach and Block Push with obstacles tasks.

skill selection. For illustrative purposes, we show the plot starting at 50k.

| Task | IRM Rand Seq | IRM CEM Seq | IRM GD Seq | Env Seq | HRL |
|---|---|---|---|---|---|
| Fetch Reach Seq | 88.1 $\pm$ 1.5 | **89.5** $\pm$ 0.34 | 86.7 $\pm$ 0.64 | 80.7 $\pm$ 4.7 | 28.4 $\pm$ 31.0 |
| Fetch Push Seq | **84.9** $\pm$ 0.12 | **84.9** $\pm$ 0.12 | 81.4 $\pm$ 1.9 | 83.7 $\pm$ 0.30 | 78.9 $\pm$ 3.1 |

Table 2: Zero-shot rewards on long-horizon manipulation tasks

## 4.3 EXTENSIONS AND ABLATIONS

**Long-Horizon Manipulation** Building on the results in Section 4.2, we demonstrate that IRM fully generalizes to solving long-horizon tasks in the setting of tabletop manipulation. During the unsupervised pretraining phase, skill discovery methods can acquire useful skills such as directional block pushing or pushing the block to certain spatial locations. We show that IRM can intelligently select a sequence of such skills to finetune via reward matching, avoiding learning a hierarchical manager policy that finetunes at the policy level.

For the Fetch Reach environment, we consider an extended horizon where the agent is tasked with reaching a sequence of goals in a particular order. For the Fetch Push task, we consider the environment depicted in Figure 2, where the agent must navigate around a barrier introduced during the finetuning phase in order to reach the goal. We compare IRM methods to an environment rollout baseline (Env Seq) and a hierarchical RL baseline (HRL). The 'IRM Seq' methods select skills based on each defined sub-task's reward function according to the IRM optimization scheme. 'Env Seq' chooses the best combination of skills based on extrinsic reward from rollouts. 'HRL' is initialized with random skills and simultaneously optimizes a manager policy over skills and the skill policies themselves. In both settings and across optimization methods, IRM outperforms the environment rollout and HRL method in identifying (Table 2) and finetuning skills (Figure 4.3 Row 2). Implementation details are provided in Appendix A.10

**Matching Metric Ablations** We validate the importance of employing the EPIC pseudometric for formulating the matching loss by ablating its contribution against more naive selections in Table 4. L1 and L2 losses are common metrics in supervised regression problems but are poor choices for comparing task similarity with rewards. Moreover, rewards can have arbitrary differences in scaling

**EPIC Loss and Extrinsic Reward are Negatively Correlated**

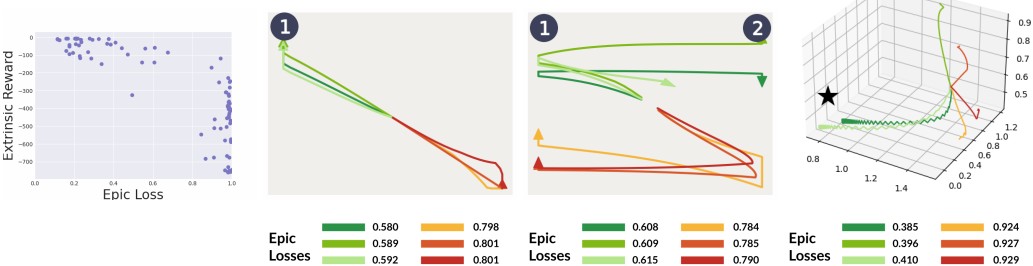

Figure 4: (a) Scatter plot of extrinsic reward vs. EPIC loss. (b) Trajectories with low and high EPIC losses for planar goal-reaching. (c) Trajectories for sequential goal-reaching. (d) Trajectories for Fetch Reach.

and shaping that L1 and L2 are not invariant to. To strengthen these comparisons, we include a learned reward scaling parameter for L1 and L2 and similarly observe that EPIC is a superior matching metric.

**Skill Discovery Algorithm Ablations** IRM is fully general to any mutual information maximization based, RL pretraining algorithm as shown in Table 3. We validate on the Fetch Reach task that IRM CEM and IRM Rand convincingly outperform all episode rollout baselines in zero-shot episode reward.

| Reward Matching | IRM CEM |
|---|---|
| IRM | **21.0** $\pm$ **0.75** |
| L1 | 8.71 $\pm$ 0.70 |
| L2 | 7.87 $\pm$ 0.86 |
| L1 + Learn Scale | 5.51 $\pm$ 1.9 |
| L2 + Learn Scale | 3.95 $\pm$ 2.2 |

Table 4: Reward matching metric ablation

## 5 ANALYSIS

**Does optimizing the EPIC loss lead to effective skill selection?** In Figure 4, we verify that EPIC loss is strongly negatively correlated with extrinsic reward on a Planar Goal Reaching task detailed in Appendix A.8. Thus, optimizing for a low EPIC loss is an effective substitute for optimizing the environment reward, and crucially, it forgoes collecting expensive environment samples.

**How can we understand skills through EPIC losses?** In Figure 5, we plot EPIC losses between intrinsic rewards and goal-reaching rewards across the 2D continuous skill space. Not only is the loss landscape smooth, which motivates optimization methods like gradient descent, but there is also a banded partitioning of the manifold. Furthermore, the latent skill space is well-structured as different darker-colored partitions of the skill space correspond to the group of skills with low EPIC loss from each task reward. EPIC losses concisely represent desirability of skills with respect

**EPIC Loss Visualizations**

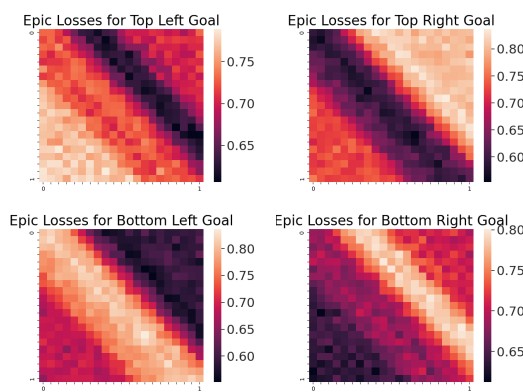

Figure 5: We examine EPIC losses between extrinsic rewards and intrinsic rewards conditioned on the skill vector. We sweep across the 2D skill vector for a pretrained planar agent.

| Intr. Rew. | IRM CEM | IRM GD | IRM Rand | Env Roll. | Env CEM | GS | Rand |
|---|---|---|---|---|---|---|---|
| DADS | **83.4** $\pm$ **2.19** | 69.9 $\pm$ 2.22 | 77.2 $\pm$ 3.83 | 74.6 $\pm$ 5.15 | 70.3 $\pm$ 5.38 | 68.9 $\pm$ 2.81 | 28.3 $\pm$ 13.5 |

Table 3: Zero-shot rewards for DADS skill discovery algorithm.

to a downstream reward function, so skills that achieve a low EPIC loss for the Top Left goal will achieve high EPIC losses for the opposite reward, Bottom Right goal.

We include a scatter plot and trajectory visualizations in Figure 4. As Figure 4 suggests, skills with the lowest EPIC loss receive high extrinsic reward, reaching the goal with high spatial precision. Skills with the highest losses produce the opposite behavior: moving in the direct opposite direction of the goal. In the sequential case, low-EPIC loss skills attempt to reach the 1st goal then the 2nd goal, while high-EPIC loss skills perform the behavior in the inverse order. The intrinsic reward module provides a much deeper insight into the semantics of skills than the extrinsic rewards obtained by skill policy rollouts.

## 6 RELATED WORK

Several works including (Sharma et al., 2020; Eysenbach et al., 2019; Achiam et al., 2018; Gregor et al., 2016b; Baumli et al., 2020; Florensa et al., 2017; Laskin et al., 2022) employ mutual information maximization for skill pretraining. While (Laskin et al., 2022) leverages coarse grid search to select skills for downstream RL, methods such as (Sharma et al., 2020) instead plan through a learned skill dynamics model at finetuning time. Our approach is similar in that it leverages pretraining model components other than the policy to guide skill selection. However, rather than generating a reward maximizing plan through possibly complex, learned environment dynamics, we instead look to match a policy to the task reward directly through a pretrained discriminator.

In the context of sequential finetuning, (Baumli et al., 2020; Eysenbach et al., 2019) employ hierarchical RL to chain pretrained skills with a manager policy requiring additional environment interactions. Works on such HRL methods include (Nachum et al., 2018; Frans et al., 2017; Vezhnevets et al., 2017; Springenberg et al., 2018) and more classically (Sutton et al., 1999; Stolle & Precup, 2002). By contrast, we demonstrate that the intrinsic reward matching framework can be extended to choose skill sequences without reliance on environment samples. The successor features line of work also adopts a unified view of skill-based RL. Such work relies on the assumption that arbitrary rewards can be parameterized linearly in some learned features and some task vector as in (Liu & Abbeel, 2021; Barreto et al., 2016). Our approach relaxes this assumption to the fully general setting by instead searching for a pretrained task with minimal proximity to an arbitrarily parameterized task reward.

## 7 DISCUSSION

We present Intrinsic Reward Matching (IRM), a framework for algorithmically unifying information maximization unsupervised reinforcement learning with downstream task adaptation. We instantiate a practical algorithm for implementing this framework and demonstrate that IRM outperforms current methods on a continuous control benchmark and tabletop manipulation tasks. IRM diverges from past works in leveraging the discriminator for downstream task inference and consequently performing skill selection without environment interactions in the short horizon setting. We also show that IRM can be readily extended to the general skill sequencing setting to solve more realistic long-horizon tasks as an alternative to hierarchical methods. Central to our contribution is a novel loss function, the EPIC loss, which serves as both a skill selection utility as well as a new way to interpret the task-level semantics of pretrained skills.

We acknowledge a number of limitations of our approach. IRM relies on samples of the state, roughly within workspace boundaries as well as access to an external reward function, ideally well-shaped, which trades off with IRM's reduced reliance on environment interactions. In order to obtain realistic image samples to compute the EPIC loss, an agent could learn an expressive generative model such as a VAE over the image states obtained during pretraining and sample from the model to generate diverse and realistic sampled states. For learning unknown state-based rewards, the agent could additionally learn an image-reward model by regressing the rewards encountered during exploration (Hafner et al., 2019). This further relaxes some of the assumptions made in this contribution and represents an exciting direction for future work.

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

# A APPENDIX

## A.1 BACKGROUND AND NOTATION

**Markov Decision Process:** The goal of reinforcement learning is to maximize cumulative reward in an uncertain environment it interacts with. The problem can be modelled as a Markov Decision Process (MDP) defined by $(\mathcal{S}, \mathcal{A}, \mathcal{P}, r, \gamma)$, where $\mathcal{S}$ is the set of states, $\mathcal{A}$ is the set of actions, $\mathcal{P}$ is the transition probability distribution, $r$ is the reward function and $\gamma$ is the discount factor.

**Unsupervised Skill Discovery:** In competence-based unsupervised RL the aim is to learn skills that generate diverse and useful behaviors (Eysenbach et al., 2019). The broad aim is to learn policies that are skill-conditioned and generalizable. Formally, we also learn skills $z \in \mathcal{Z}$ and take actions according to $a \sim \pi(\cdot|s, z)$. As an illustrative example, applying this formalism to the Mujoco Walker domain, we might hope to find a skill-conditioned policy and skills $z_{\text{walk}}, z_{\text{run}}$ such that $\pi(\cdot|s, z_{\text{walk}})$ makes the agent walk, while $\pi(\cdot|s, z_{\text{run}})$ makes it run. Further, if we allow for continuous skills, we can also imagine being able to use the policy to "jog" at different speeds by interpolation the $z_{\text{walk}}$ and $z_{\text{run}}$ skills. That is, taking $z_{\text{jog}}^{\alpha} = \alpha \cdot z_{\text{walk}} + (1 - \alpha) \cdot z_{\text{run}}$ should, intuitively, yield a policy $\pi(\cdot|s, z_{\text{jog}}^{\alpha})$ that makes the agent jog at speed dictated by the parameter $\alpha$.

**Finetuning Pretrained Skills:** With a skill-conditioned policy $\pi(\cdot|s, z)$, an agent needs to infer which skill to index for a downstream task (e.g. identifying if it needs to use $z_{\text{walk}}$ or $z_{\text{run}}$) during finetuning. This is a relatively under-explored area, with the most universal approach being a coarse, discretized grid search. Least squares regression has also been investigated in the context of successor features (Liu & Abbeel, 2021).

## A.2 COMPETENCE-BASED SKILL DISCOVERY

Competence-based skill discovery algorithms aim to maximize the mutual information between trajectories and skills:

$$I(\tau; z) = \mathcal{H}(z) - \mathcal{H}(z|\tau) = \mathcal{H}(\tau) - \mathcal{H}(\tau|z) \tag{7}$$

Since the mutual information $I(s; z)$ is intractable to calculate in practice, competence-based methods maximize a variational lower bound. Many mutual information maximization algorithms, such as Variational Intrinsic Control (Gregor et al., 2016a) and Diversity is All You Need (Eysenbach et al., 2018), use the estimate $I(\tau; z) = \mathcal{H}(z) - \mathcal{H}(z|\tau)$. Other competence-based methods, such as Dynamics-Aware Unsupervised Discovery of Skills (Sharma et al., 2019), Active Pretraining with Successor Features (Liu & Abbeel, 2021), and Contrastive Intrinsic Control (CIC) (Laskin et al., 2022), maximize a lower bound for $\mathcal{H}(\tau) - \mathcal{H}(\tau|z)$.

While the decompositions of the mutual information objective are equivalent, algorithms make different design choices regarding how to approximate entropy, represent trajectories, and embed skills. These choices affect the distillation of skills: for instance, without explicit maximization of $\mathcal{H}(\tau)$ in the decomposition of mutual information, behavioral diversity may not be guaranteed when the state space is much larger than the skill space (Laskin et al., 2022).

## A.3 CIC

Contrastive Intrinsic Control (CIC) (Laskin et al., 2022) is a state of the art algorithm for competence-based skill discovery. CIC maximizes a lower bound for $I(\tau; z) = \mathcal{H}(\tau) - \mathcal{H}(\tau|z)$ through a particle estimator for $\mathcal{H}(\tau)$ and a contrastive loss from Contrastive Predictive Coding (CPC) (van den Oord et al., 2019) for $\mathcal{H}(\tau|z)$. The lower bound for $I(\tau; z)$ is:

$$I(\tau; z) \geq F_{\text{CIC}}(\tau; z) := \mathcal{H}_{\text{particle}}(\tau_i) + \mathbb{E}\left[ q_\phi(\tau_i, z_i) - \log \frac{1}{N} \sum_{j=1}^{N} \exp(q_\phi(\tau_j, z_i)) \right] \tag{8}$$

where $\mathcal{H}_{\text{particle}}(\tau) \propto \sum_{i=1}^{n} \log ||h_i - h_i^*||$, $h_i^*$ is the k-Nearest Neighbors embedding, $N_k$ is the number of k-NNs used to approximate entropy, and $N - 1$ is the number of negative samples.

## A.4 DADS

We additionally use Dynamics-Aware Unsupervised Discovery of Skills(DADS) (Sharma et al., 2020) for skill discovery, as it is one of the few skill discovery algorithms to successfully scale up to continuous skills. DADS maximizes a lower bound for $I(\tau; z) = \mathcal{H}(\tau) - \mathcal{H}(\tau|z)$ through learning skill-conditioned transition distributions. The lower bound for $I(\tau; z)$ is:

$$I(\tau; z) \geq F_{\text{DADS}}(\tau; z) := \log \frac{q_\phi(s'|s, z)}{\sum_{i=1}^{L} q_\phi(s'|s, z_i)} + \log L \qquad (9)$$

For our experiments, we reimplement the on-policy DADS algorithm in PyTorch. We follow the default hyperparameters and train for 20 million environment steps, per (Sharma et al., 2020).

## A.5 ENVIRONMENT DETAILS

The URLB domains are Walker, Quadruped, and Jaco. Walker requires a bipedal agent to perform a variety of navigation based tasks on a 2D-plane while preserving its balance. Quadruped, a more challenging domain due to a higher-dimensional state-action space, requires a quadrupedal agent to perform navigation tasks in a 3D environment. Jaco robot arm is a 6-DOF manipulator with a three-finger gripper which contains a variety of directional reaching tasks

For URLB (Laskin et al., 2021) environments, we follow default environment settings. Like many skill-discovery methods (Sharma et al., 2020) (Eysenbach et al., 2019), we restrict the discriminator input. For Quadruped, we use the $x, y, z$ velocity, which is included in the environment's state space. For Walker, we use the $x, y, z$ world-position, which we add to the environment's state space but remove from the policy input. For Jaco, we use the $x, y, z$ world position.

For our fetch reaching environment, we use the Gym Robotics Fetch environment (Brockman et al., 2016). We set the time limit to 200. For the fetch push environment, we partition the continuous action space into 4 actions, which involve pushing the block forward, backward, left, and right. We set the time limit to 10 for skill learning.

We evaluate sequential skill selection on 2 environments: Fetch Reach and Fetch Push. For the Fetch Push task, we fix 3 waypoints, depicted in Figure 2 and fix a time horizon of 15 pushes per waypoint. For Fetch Reach, we consider 2 waypoints and a time horizon of 25 for each waypoint.

Our plane environment is a 2D world with observations in [-128, 128] x [-128, 128] and continuous actions in [-10, 10] x [-10, 10].

## A.6 PRETRAINING HYPERPARAMETERS

For the Jaco domain we use a skill dimension of 2 and a discriminator MLP hidden dimension of 64. We use an alpha value of 0 for the entropy weighting as in (Laskin et al., 2022). We input the 3D position of the end-effector of the Jaco arm to the discriminator. For the Walker domain we use a skill dimension of 2 and a discriminator MLP hidden dimension of 256. We use an alpha value of 0.7 for the entropy weighting. We input the displacement in the 3D position of the torso of the walker to the discriminator. For the Quadruped domain we use a skill dimension of 16 and a discriminator MLP hidden dimension of 128. We use an alpha value of 0.5 for the entropy weighting. We input the 3D velocity of the body of the quadruped to the discriminator. We use a learning rate of 1e-4, a critic target tau parameter of 0.01, and a constant standard deviation exploration schedule of 0.2. The rest of the RL hyperparameters are as in (Laskin et al., 2021).

For the Fetch Push environment, we use a skill dimension of 16 and a discriminator MLP hidden dimension of 16. We use an alpha value of 0 for entropy weighting. For the Fetch Reach environment, we use a skill dimension of 8 and a discriminator MLP hidden dimension of 64. We use an alpha value of 0 for entropy weighting. For all environments, we use a replay buffer size of 100k.

## A.7 INTRINSIC REWARD MATCHING AND ENVIRONMENT ROLLOUT BASELINE HYPERPARAMETERS

IRM CEM and Env Rollout CEM are trained for 5 iterations with 1000 samples at each iteration and 100 elites selected each iteration. Env Rollout CEM consumes the entire downstream finetuning

**Zero-Shot Performance for Planar Goal Reaching**

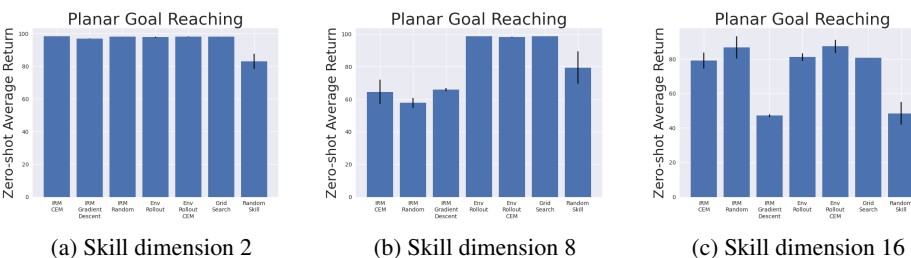

(a) Skill dimension 2   (b) Skill dimension 8   (c) Skill dimension 16

Figure 6: Zero-Shot Returns for Planar Goal Reaching averaged over 5 seeds

budget on just skill selection. For illustrative purposes, we start its plot at 50k steps to show that finetuning still occurs, however, sample-inefficiency suffers due to excessive rollouts for skill selection. This problem only worsens for long time horizons. IRM Gradient Descent is trained for 5000 steps with a learning rate of 5e-3 and initialized at the skill vector of all 0.5s. IRM Random selects 100 random skills. Env Rollout trials 10 random skills for a fully episode. Grid Search coarsely trials 10 skills from the skill of all 0s to the skill of all 1s as in (Laskin et al., 2021).

### A.8   Planar Goal Reaching

The planar goal reaching task consists of a simple 2D plane with a point with a 2D Cartesian state space that can displace in the x and y coordinates with a 2D action space. Skills learned tend to span the 2D space reaching to diverse locations distributed broadly across the environment. We show some sample zero-shot skill selection results over three different skill dimensions in Figure 6.

### A.9   Finetuning Performance on URLB

In Figure 7 we compare the finetuning sample-efficiency of IRM methods against environment rollout-baselines on the URLB Walker tasks. IRM performs skill selection with 0 environment interactions. The episode length of the URLB environments is 1000, meaning that in order to evaluate a single skill, rollout based methods must exhaust 1000 environment steps (i.e. grid search spends 1000 * 10 = 10,000 environment steps - 10 percent of the available finetuning budget). By contrast, our method immediately uses new environment steps for improving the policy. As a result, the IRM based approaches generally achieve greater sample efficiency, even when initial skill selection obtains similar performance to the rollout based methods. For illustrative purposes we have shown Env CEM starting at 50k steps even though it far exceeds the 100k sample budget to select a skill before making any RL updates due to having to execute full episode rollouts in the inner loop of optimization. This issue worsens with increasing episode lengths. We plot results over 3 seeds with standard error shading.

### A.10   Sequential Skill Selection

For sequential skill selection, we compare IRM Sequential and Environment Sequential skill selection. IRM Sequential consists of an iterative process. The first skill is chosen entirely free of environment samples, exactly identical to the single-skill tasks. Once the first skill is chosen, we roll out a trajectory with the skills we have chosen so far and use the latter half of the trajectory as the Pearson samples for our EPIC loss. We use Gaussian noise with variance 1 for our Canonical samples as described in Appendix A.12.2. At each step of the skill selection process, we use the corresponding IRM optimization methods.

For our Environment Sequential skill selection method, we select skills iteratively as well. For each waypoint or subtask, we randomly sample $N$ skills and commit to the best, where $N = 10/n\_subtasks$.

**Finetuning Performance on URLB**

Figure 7: IRM finetuning results compared to rollout-based baselines on Walker URLB tasks.

| Skill Dim | IRM CEM | IRM GD | IRM Rand | Env Roll. | Env CEM | GS | Rand |
|---|---|---|---|---|---|---|---|
| 8 | $21.1 \pm 0.51$ | $15.7 \pm 1.61$ | $18.9 \pm 0.18$ | $18.4 \pm 0.18$ | $18.8 \pm 0.48$ | $17.9 \pm 0.101$ | $13.5 \pm 1.85$ |
| 16 | $17.4 \pm 1.30$ | $14.6 \pm 0.63$ | $18.8 \pm 0.26$ | $22.7 \pm 0.83$ | $23.1 \pm 0.36$ | $14.0 \pm 0.19$ | $11.2 \pm 2.32$ |
| 32 | $20.1 \pm 0.54$ | $22.537 \pm 0.25$ | $19.8 \pm 0.14$ | $22.2 \pm 0.58$ | $21.5 \pm 0.67$ | $24.0 \pm 0.12$ | $19.9 \pm 0.67$ |
| 64 | $21.9 \pm 0.48$ | $1.68 \pm 0.069$ | $20.9 \pm 0.74$ | $22.5 \pm 0.70$ | $21.6 \pm 0.89$ | $18.2 \pm 0.059$ | $13.3 \pm 2.15$ |

Table 5: IRM methods and environment rollout methods ablated over multiple skill dimensions on Fetch Push

## A.11 HIERARCHICAL REINFORCEMENT LEARNING BASELINE

In order to validate the benefits of IRM's offline skill selection, we compare against a baseline that leverages a conventional hierarchical RL algorithm to solve long-horizon, sequential tasks. We instantiate a TD3 manager agent that outputs into a skill action space from state input at a temporally abstract timescale. As in the IRM setup, this timescale is fixed to align with the changes in reward to encourage the manager to change its skill prediction according to the change in the reward semantics. The manager's is then inputted to the low-level pretrained skill policy which is rolled out over many steps with the skill fixed. Both the manager policy and the low-level policy weights are updated during finetuning. The manager agent is randomly initialized such that its initial skill prediction is random.

## A.12 ADDITIONAL ABLATIONS

### A.12.1 SKILL DIMENSION

We ablate skill dimension and evaluate the zero-shot performance of all skill selection methods. IRM's performance generally increases with increased skill dimension despite discriminator over-fitting issues associated with larger skill spaces. The IRM GD learning rate is chosen as 5e-3 for all experiments in this work and is not tuned at all. Such likely explains the divergence of the 64 dimensional result.

### A.12.2 PEARSON & CANONICAL DISTRIBUTIONS

We experiment with many ways to approximate the Pearson and Canonical distributions. We defined Full Random to be our uniform samples from a reasonable estimate of the upper and lower bounds

for each dimension of the state. For our planar environment, the bounds are defined explicitly and thus known; for more complex environments, we estimate the bounds. For example, for a tabletop manipulation workspace, we sample 2-dimensional block positions uniformly within the rectangular plane of the table surface. In practice, IRM is fairly robust to the distributions, though there are subtleties that emerge in the various choices for the Pearson and Canonical distributions. For instance, we also ablate a Uniform(0,1) distribution, which generally performs much worse, due to lack of state coverage for most environments. For the Canonical distribution, we also approximate samples by perturbing the Pearson samples by $\epsilon$ sampled from a Gaussian distribution. We experiment with hyperparameters of variance, which may be adjusted based on the environment. For our sequential IRM method, we use this Canonical distribution to ablate on-policy samples.

| Pearson Distribution | Canonical Distribution | IRM CEM |
|---|---|---|
| Full Random | Full Random | $20.341 _{\pm 0.306}$ |
| Full Random | Uniform(0,1) | $16.343 _{\pm 0.708}$ |
| Full Random | $\epsilon \sim \mathcal{N}(0, 1)$ | $21.191 _{\pm 0.629}$ |
| Full Random | $\epsilon \sim \mathcal{N}(0, 0.1)$ | $21.027 _{\pm 0.419}$ |
| Uniform(0,1) | $\epsilon \sim \mathcal{N}(0, 1)$ | $5.905 _{\pm 3.157}$ |
| Uniform(0,1) | $\epsilon \sim \mathcal{N}(0, 0.1)$ | $2.851 _{\pm 0.605}$ |

Table 6: EPIC Loss Sampling Distribution Ablations.

None of the distributions ablated above require on-policy environment samples. It *is* possible to use on-policy samples for the state distributions, and we choose to do so for our sequential IRM method, as previous skill rollouts may provide useful Pearson samples for the subsequent skill selection. Note that while on-policy Canonical samples are possible, they are incredibly expensive and require access to the environment simulator, so we focus on other choices of distributions.

### A.12.3  SPARSE REWARD ABLATION

We ablate our planar EPIC Loss visualizations with sparse rewards. Instead of a well-shaped goal-reaching reward, we use sparse rewards based on the tolerance to the goal. We define the tolerance as the radius the agent must be within if our 2d planar environment is scaled to [0, 1] x [0, 1]. With a very sparse reward, we show that EPIC losses are largely uninformative. However, by slightly relaxing the tolerance, we show a much better shaped EPIC loss landscape that bears similarity to that of Figure 5. Thus, while our method is dependent on access to extrinsic rewards, and ideally, shaped rewards, we show that the EPIC loss landscape over sparse reward landscapes with sufficient tolerance can be optimized.

**EPIC Loss Visualizations**

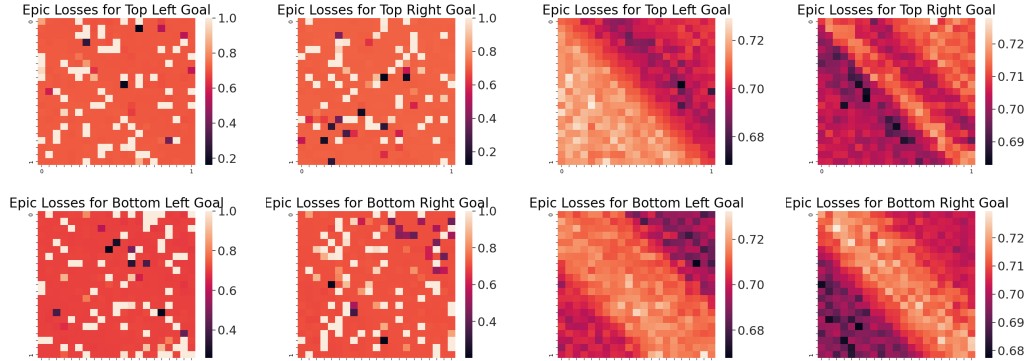

Figure 8: We examine EPIC losses between extrinsic rewards and intrinsic rewards conditioned on the skill vector. We sweep across the 2D skill vector for a pretrained planar agent. Left: Sparse goal-reaching reward with tolerance of 0.03. Right: Sparse goal-reaching reward with tolerance of of 0.07.

