# OpenReview forum: "Skill-Based Reinforcement Learning with Intrinsic Reward Matching"
_ICLR.cc/2023/Conference — Submitted to ICLR 2023_

### Official Review · Reviewer_jnVw · 2022-10-23

**Confidence:** 4
**Correctness:** 3
**Technical Novelty And Significance:** 4
**Empirical Novelty And Significance:** 3
**Recommendation:** 6

**Clarity, Quality, Novelty And Reproducibility:**

As described above, while I like the idea and think it is an interesting view of some skill based learning methods I think the clarity and quality of the paper would need to be improved for it to meet the bar for this venue.

**Strength And Weaknesses:**

I think the idea and method being proposed in this paper are somewhat interesting and, to my knowledge, novel. However, apart from some methodological issues I have with the idea, my main concern is the quality of writing and communication in the paper. There were multiple instances when reading the paper which made me think that the work was quite rushed and for which I had to spend a lot of time looking through the Appendices in order to make sure I understood the details. While I am willing to ignore some of these, when they repeat so often it makes the paper much less enjoyable to read and, more pertinent to the context of reviewing, much harder to parse and understand.

I will first list out some questions I have regarding the method and will then follow with a list of ways in which I think the communication could be improved, including areas where I think claims need to be backed up with more references to the literature. I’d also like to note that if the writing is improved, I would be more than happy to reconsider my score.

Methodological questions:
1. In the Appendix, the choices of D_S and D_A are ablated in Table 6. There seems to be quite a stark difference when switching from the ‘Uniform’ distribution for the Pearson distribution to the ‘Full Random’. My understanding was the ‘Full Random’ samples uniformly but with the upper and lower bound for each state fixed. It was not clear to me why changing the bounds would affect the sampling distribution so much unless it was quite a strong assumption. Could the authors clarify what these two distributions mean perhaps with an example?

2. Unfortunately I’m not convinced the method works particularly well given the results of Table 1 and Figure 3. In Table 1, for the JACO task the Env CEM baseline seems to sometimes get a reasonably higher performance (9-10 on Jaco Top and Bottom Right). In the Walker setting a ‘Random’ skill works so well that I am genuinely a bit perplexed as to why other methods are so terrible - for instance in Walker Walk. In the finetuning results of Fetch Reach as well I would not say that IRM is that much more sample efficient. Perhaps understanding this a bit better - for instance why the method does not work so well on Walker could lead to algorithmic improvements that make it more robust. As things stand I don’t think the empirical evidence is compelling enough for this to be used in practice.
The Jaco tasks also bring up an interesting point regarding sparse reward tasks. Sparse rewards are easier to define but as the authors note, would be much harder to use as a way to pick skills via matching. In general perhaps the method is sensitive to the nature of the reward landscape? In particular how does the analysis of Figure 5 of the EPIC landscape change for different tasks? Currently I’m not certain if the performance is limited because R_int is not being estimated accurately enough or if the loss landscape itself is not smooth on some tasks and understanding this would be quite interesting I think.

3. In Figure 3, why would we expect the finetuning performance to be different under IRM?  My understanding is that the method would try to pick the ‘most suitable skill’ in terms of how it matches with the task reward. This says nothing about the skills ability to finetune on a new task per se, does it?

4. In Section 7, the EPIC loss is described as a contribution of this work. My understanding was that this loss was used, albeit in a different context, in prior work?

Areas in which presentation can be improved and clarified:
1. There are multiple places (Section 1, 2.1) with sentences in the vein of - “the intrinsic reward function learned during skill pre-training can..”. First, this notion that when using a particular formulation to learn skills, we can interpret the skill as optimizing an intrinsic reward function is somewhat new. This idea needs to be introduced and at least some context given to the reader to understand what it means. Also I don’t think the claims can be made generically across *all* skill learning methods. At least a gentle introduction of what skill learning methods this applies to would make this part much more readable. Often these sentences do not have references and citations to specific works to back them up as well which is an issue in scientific writing.

2. The equations relating to the EPIC losses - Equations 2 and 3 are hard to parse and not strictly correct I think. Equation 2 includes S, A, S’ on the right-hand-side which should be input to the function. The definition in Equation 3 is also inconsistent with that of (2) and trajectories are defined as (s, s’) which means the action is not used. I understand that the reward functions being considered here do not require actions but this should be specified explicitly in the main text and in either case the mathematical definitions should be consistent.

3. The second point is particularly important in the context of Algorithm 1 since these equations are the only way to understand it. In general, as far as possible, algorithm boxes should be self-contained and express the main jist of the approach. Currently in the box variables N_FT, S_p, A_p, S’_p, S_c, A_c, S’_c are defined but never used and variables s and s’ are used but not properly introduced. I think if the algorithm box just explicitly wrote out the EPIC loss from Equation 2 it would make it much more self contained and easier to parse.

List of typos and other less minor issues in writing:
1. Paragraph 3, Introduction: paramterized instead of parameterized.

2. Section 2.1 after Equation (1): ‘is’ is missing after ‘where VLB<= I(\Tau, z)’.

3. Equation (3) and (4) - it equation does not include over what random variables the expectation is being taken.

4. Section 3.1 paragraph 1: ‘specified our intrinsic’ should be ‘specified by our intrinsic’ and ‘As is such’ should be ‘As such’.

5. Section 3.1, paragraph 2: ‘which our pretrained’ should be ‘which of our pretrained’.

6. Section 3.2, paragraph 3: ‘leads improved’ should be ‘leads to improved’.

7. Table 6 v/s 6, Appendix 1 v/s 1 etc. are used interchangeably throughout. It helps when the relevant sections and appendices are explicitly and consistently labeled  throughout the text. As a specific example, the ablation for P_s and P_A at the end of section 3.2 refer to ‘6’ which is unclear.

8. Section 4.1, paragraph 1: ‘is either comparable to our outperforms..’ - ‘our’ should be ‘or’.

9. Section 4.3: ‘We compare an extended versions’ - no ‘on’. Sentence ends with ‘...environment rollout method identifying’. Figure 4.3 Row 2 should be Figure 3?


EDIT:
After looking through the changes made during the rebuttal period I think the paper reads much much more clearly and is stronger now. There are still some nagging issues in the presentation (predominantly of Section 3) but I appreciate that the authors also ran experiments for other baselines and it is challenging to satisfy all constraints within a short time limit. I have increased my scores to reflect the updated paper and would like to thank the authors for the great work in addressing concerns from the rebuttal period.


**Summary Of The Paper:**

This paper presents a method that learns to choose among a set of skills to solve a downstream task. The key idea presented is that some skill learning methods often learn a discriminator which can be thought of as an intrinsic reward function for the underlying skill. Using an equivalent policy-invariant metric called EPIC, the authors suggest comparing the ‘fitness’ of skills using the corresponding intrinsic rewards to the external task-specific environment reward. This presents a zero-shot method to find a good skill for an unseen task. This method is tested on a number of benchmarks including one that involves composing skills sequentially and the intrinsic reward and EPIC loss are also analyzed.

**Summary Of The Review:**

As things stand, I think the paper has too many flaws for it to be published at ICLR. I am happy to reconsider though based on the rebuttal.

---

> ### Author Response · Authors · 2022-11-09
> **Addressing methodological questions 1 through 3**
>
> Thank you for raising several points of improvement particularly regarding fluency. We largely appreciate the merit of these suggestions and believe that many of them can be very readily integrated.
>
> **Sampling Distributions**
>
> You are correct in observing that both “Full Random” and “Uniform” are uniform distributions, albeit Uniform with bounds between 0 and 1 and full random within the workspace bounds. We will revise the table to make this distinction clear. Defining workspace bounds is often a part of environment design and thus we see this as a rather weak assumption. In truth, these bounds need only be approximate, the main key being that samples roughly cover the space of realistic samples the agent may encounter. The insight to be gleaned is that sampling in too small a radius (between 0 and 1) has a detrimental effect on the performance of the method as samples will not reflect this realistic agent state visitation distribution. Much of this discussion has more to do with the specifics of the EPIC metric computation than our core contribution, which is why we have deferred details to the appendix.
>
> **Experimental Settings and Empirical Novelty**
>
> On the latter note of sparse rewards, we agree as explained in the paper that such a setting might be more challenging for matching based approaches due to the discontinuous nature of the reward function. Rollout based approaches also inherit challenges in skill selection as they are reliant on a rolled-out skill fully solving the task. For simpler tasks like Jaco this may work at times, however, in general, more flexible sampling strategies such as sampling positions uniformly at random in the workspace present a more promising methodology towards extracting signal in sparse settings. This is because we can practically scale up the number of offline samples to get greater coverage and increase the likelihood of sampling rewarded states, whereas, we are constrained to our skill policy’s state visitation in order to get sparse signal when using rollout-based methods. Moreover, as suggested, an additional analysis into the EPIC landscape for such settings would indeed be interesting - we will look into providing this before the conclusion of the rebuttal period.
>
> Current skill pretraining algorithms do not yet consistently provide a sufficiently rich skill library to enable positive transfer on every task in URLB and thus the full potential of the gains from optimal skill selection on sample-efficiency are yet to be realized. Particularly for the Walker environment, which is the most challenging of the continuous control environments due to its instability, most of the learned skills are very far from having positive transfer to the tasks with the exception of standing. For these settings, choosing a skill optimally is just as good as random. However, as skill pretraining methods improve on these more complex tasks, the performance gaps between skills will increase as will the premium on finetuning samples further increasing the appeal of interaction-free skill selection methods. We hope that principled finetuning algorithms like IRM will encourage further improvements to unsupervised RL pretraining methods.
>
> Furthermore, IRM performs skill selection with 0 environment interactions. The episode length of the URLB environments is 1000, meaning that in order to evaluate a single skill, rollout based methods must exhaust 1000 environment steps (i.e. grid search spends 1000 * 10 = 10,000 environment steps - 10% of the available finetuning budget). By contrast, our method immediately uses new environment steps for improving the policy. We will correct our omission to shift all of the rollout based methods to the right in Figure 7.
>
> **Finetuning Efficiency**
>
> You are technically correct - finetuning performance is enhanced only to the degree that the zero-shot performance of the selected skill is enhanced. We view both skill selection and skill finetuning to task reward as part of the overall finetuning process, hence the choice of language. However, we will adjust our language to make clear that we are concerned with zero-shot performance.

---

> ### Author Response · Authors · 2022-11-09
> **Addressing 4th methodological question and insightful suggestions on clarity**
>
> **EPIC Loss Novelty**
>
> To our knowledge, EPIC loss has not been presented in prior work. The EPIC reward comparison metric is presented in Gleave et al. 2020 in the context of comparing hand-designed reward functions in very simple gridworld environments with low-dimensional state and action spaces. Prior works do not
> - Formulate the EPIC metric as a differentiable loss function that is minimized to find an optimal policy
> - Use the EPIC metric to compare reward functions that are learned as a part of an unsupervised learning process
> - Approximate the EPIC metric by selecting bespoke distributions for the expectation in the canonicalization and the Pearson correlation
> - Analyze the EPIC manifold to make claims about smoothness and partitioning in skill space
>
> These investigations, amongst others, are central to the novelty of our approach.
>
> **Background on Skill Pretraining**
>
> We provide the mutual information variational lower bound in an attempt to concisely unify skill pretraining without belaboring details from prior work. However, you are correct in observing that skill discovery is a relatively nascent literature and thus we could benefit from some brief contextualization. We will look into this and follow-up with a revision. Furthermore, we are happy to weaken our claim that this applies universally for all skill pretraining methods as well as provide additional references to justify claims in the earlier sections of the paper.
>
> **Action-Independent Reward Notation**
>
> Since you are correct in observing that the rewards in our setting are not action dependent, we will revise our notation in Section 2.2 for consistency as suggested.
>
> **Self-Contained Algorithm Presentation**
>
> We agree that, at the expense of some brevity, we should be more explicit in Algorithm Box 1 about how the Pearson samples are used to compute the EPIC distance. We will make these adjustments before the conclusion of the rebuttal period.
>
> We will correct the notes typographical errors as well as work on the overall fluency of the writing for easier readability.

---

> > ### Comment · Reviewer_jnVw · 2022-11-15
> > **Brief reply to comments**
> >
> > Thanks for the detailed reply. I took a brief look at the comments but will wait for the updated paper before updating my score - as I said, the communication was quite a big problem for me in the original submission. Having said that if the authors do follow up with the points mentioned here, I see no reason not to increase my score.
> >
> > One point in the rebuttal that I did want to address now, since it could also be relevant to the changes being made to the paper, is this sentence in the rebuttal: "However, you are correct in observing that skill discovery is a relatively nascent literature and thus we could benefit from some brief contextualization." I did not mean to suggest that skill discovery is relatively nascent at all this at all! My point in fact is quite the opposite - this is a field that has been studied for quite some time. I have listed some papers below to drive this point home although even this list is far from exhaustive and really just barely scratches the surface. Note however that the SkillS paper from Thrun dates back to 1994. Their results are on grid-worlds and I don't mean to suggest that these methods should be considered state-of-the-art today nor am I even asking that they all be cited in the paper. I think it is important for the paper to situate itself more clearly in skill learning literature because it helps the reader have some context into the algorithm especially given there are so many relevant ideas in the space.
> >
> > My point is this : The ideas described here are interesting but assume certain kinds of skill discovery methods that should be properly clarified in the introduction. I hope the updated paper addresses this!
> >
> > I also think this point is not very strong:
> >
> > "Current skill pretraining algorithms do not yet consistently provide a sufficiently rich skill library to enable positive transfer on every task in  URLB and thus the full potential of the gains from optimal skill selection on sample-efficiency are yet to be realized. Particularly for the Walker environment, which is the most challenging of the continuous control environments due to its instability, most of the learned skills are very far from having positive transfer to the tasks with the exception of standing. For these settings, choosing a skill optimally is just as good as random. However, as skill pretraining methods improve on these more complex tasks, the performance gaps between skills will increase as will the premium on finetuning samples further increasing the appeal of interaction-free skill selection methods."
> >
> > The way I read this is: It requires me to assume that in future some skill pretraining methods will work really well, fit into the description proposed by the paper and then we will see benefits of using it on the Walker task? Sorry it just feels like a strange argument to make. I understand where you are coming from, but unfortunately I will have to evaluate the method based on the empirical evidence provided with the current skill based methods. However the argument regarding the additional cost of rollout based methods is more fair and should be highlighted in the paper as well. I look forward to seeing the new plots to base my decision on.
> >
> > Thanks again for the replies!
> >
> > [1] Thrun, Sebastian, and Anton Schwartz. "Finding structure in reinforcement learning." Advances in neural information processing systems 7 (1994).
> > [2] M. Pickett and A. G. Barto. Policyblocks: An algorithm for creating useful macro-actions in reinforcement
> > learning. In ICML, volume 19, pages 506–513, 2002.
> > [3] Ranchod, Pravesh, Benjamin Rosman, and George Konidaris. "Nonparametric bayesian reward segmentation for skill discovery using inverse reinforcement learning." 2015 IEEE/RSJ International Conference on Intelligent Robots and Systems (IROS). IEEE, 2015.
> > [4] Shiarlis, Kyriacos et al. “TACO: Learning Task Decomposition via Temporal Alignment for Control.” ICML (2018).

---

> > > ### Author Response · Authors · 2022-11-17
> > > **Feedback Incorporated in Paper Revision**
> > >
> > > Thank you again for your suggestions and comments! We have restructured and rewritten many sections of our paper to incorporate your feedback. We add additional quantitative results and hope this provides a fuller empirical understanding of IRM.
> > > 1. We clarify our **Pearson and Canonical sampling distributions** in Table 6.
> > > 2. We improve our notation, clarity, and wording of **Section 2.2: Equivalent-Policy Invariant Comparison**. We hope this restructuring clarifies any outstanding notational concerns.
> > > 3. We have reformatted and rewritten our **algorithm**. We hope this clarifies any confusion of notation in regards to Section 2.2.
> > > 4. We have restructured and clarified our **Section 2.1 Unsupervised Skill Pretraining** to include a more comprehensive background on skill discovery.
> > > 5. We correct typos, grammatical, and other writing errors throughout the paper. Thank you for pointing these out, and we hope these improvements reduce confusion and improve fluidity!

---

### Official Review · Reviewer_1F21 · 2022-10-24

**Confidence:** 4
**Correctness:** 3
**Technical Novelty And Significance:** 3
**Empirical Novelty And Significance:** 2
**Recommendation:** 6

**Clarity, Quality, Novelty And Reproducibility:**

- The presentation is basically clear in general, but please check out my comment above.
- The overall quality of the paper is fine.
- I think their approach to matching intrinsic and extrinsic reward functions using the skill discriminator and the pseudometric for reward functions for skill selection is somewhat novel.

**Strength And Weaknesses:**

Strengths

- The idea of using distances between intrinsic reward functions induced by the skill discriminator and extrinsic reward functions in downstream tasks to choose one of the pre-trained skills is novel to some degree.
- Experimentation with the proposed approach is thoroughly done, which provides the empirical evaluation and analysis of the method in multiple aspects.

Weaknesses

- If I'm not mistaken, one important weakness of this work is that it requires more information about the environment and downstream tasks, compared to most prior skill discovery approaches, which makes the underlying problem settings different. The computation of the EPIC loss requires the ranges of state dimensions (or the state distribution) and the extrinsic (i.e., downstream task) reward functions, which, especially the latter, are usually not assumed by prior approaches.
- In the writing of the paper, the authors should be clear about the distinction in the problem settings. They mention that this approach can "determine the optimal skill for an unseen task without environment samples" and is sample-efficient (in the abstract, Sec.3.2, Sec.4.2, etc.), but determining the extrinsic reward function without given knowledge requires or even may not be easily possible (e.g., in sparse-reward environments) with samples from environment interactions.
- The writing could be improved in some minor ways. For instance, I believe "invariant on an equivalence class of reward functions that always induce the same optimal policy" in Sec.2.2 is quoted from the EPIC paper (Gleave et al. (2020)) as-is. Both "fine-tune" and "finetuning" appear in the same Algorithm 1.

**Summary Of The Paper:**

In the field of unsupervised skill discovery, the authors focus on that the skill discriminator is often used for generating intrinsic reward signals and propose to leverage the skill discriminator to match the intrinsic rewards with extrinsic rewards to solve downstream tasks. They use the EPIC pseudometric to measure distances between intrinsic and extrinsic reward functions, and test a number of methods for minimizing the distances and choosing optimal skills. The empirical evaluation and analysis are done in URLB and the Fetch environment.

**Summary Of The Review:**

I like the idea of exploiting the skill discriminator and the reward pseudometric to select optimal skills for downstream tasks and the experiments and analyses presented in the paper. On the other hand, as I mentioned in the Strength And Weaknesses section, to my understanding, it requires the additional information. I believe it would be fairer to state such additional assumptions or difference in the problem settings, because the knowledge of extrinsic reward functions on new tasks is not necessarily cheaper than environment interactions.

---

> ### Author Response · Authors · 2022-11-09
> **Some comments on design assumptions and contribution scope**
>
> Thank you for raising some important questions regarding design assumptions made in our work. We are integrating suggestions made around making certain details more explicit and wanted to leave a few brief remarks regarding the intended scope our our contribution.
>
> **Assumed Additional Information**
>
> While you are correct to point out that our work relies on some additional information, we do not see the underlying problem setting as fundamentally different from general skill-based RL. Defining workspace bounds is often a part of environment design - many RL pipelines use rough workspace bounds in defining the observation space, yet need not sample from this space and thus do not make mention of this prior information. Furthermore, in our case these bounds need only be approximate, the main key being that samples roughly cover the space of realistic samples the agent may encounter. In principle one could further relax these assumptions by sampling from a learned generative model over states encountered during pretraining, however, we saw this complexity as unnecessary given the above points. You are correct in that our method does require access to the task reward function, as we state in section 3.1. That being said, we will look into making these details more explicit.
>
> **Availability of Reward Functions**
>
> In practice, experimental designers often implement handcrafted task reward functions in order to encourage learning a desired behavior as a part of the task and environment design. Whereas there are certainly instances in which this is not the case, or the reward is not computable from available state information, we do not focus our contribution on these scenarios, leaving an extension of IRM to the task reward learning setting for future work.
>
> **Clarity in Formal Statement of EPIC Metric**
>
> We look to make the formal statement of the EPIC metric more clear in Section 2.2 as well as improve overall fluency and fix typographical errors.
>
> Overall, we believe that most of the design assumptions made in the contribution are relatively mild and do little to affect the general problem setting. Our aim was to produce a principled algorithmic framework for connecting skill pretraining to downstream finetuning, and we hope that our work will encourage further extensions such as reward learning integration as you remarked. However, we will certainly look into making design assumptions and their implications on sample-efficiency more explicit for enhanced clarity before the conclusion of the rebuttal period.

---

> > ### Comment · Reviewer_1F21 · 2022-11-16
> > **Response to Authors**
> >
> > Thanks for the response from the authors.
> >
> > On the assumptions about the additional information, while I think that the proposed algorithm possesses its own value, I don't believe such assumptions are mild. Although it is not very rare for experimental designers to define task reward functions, in order to tackle real-world problems, especially when it comes to one of the main goals of skill discovery, the application of learned skills to solve "novel" downstream tasks, about which even humans may not have full knowledge, the assumption about the availability of reward functions doesn't seem "mild". This is also why I think stating that it can "determine the optimal skill for an unseen task without environment samples" without being clear about such assumptions is not a very fair way of presenting the contributions.
> >
> > Please note that using almost a sentence from a previous literature as-is is about more than clarity or fluency, and I hope any concerns like that get addressed in the future versions of this work.

---

> ### Author Response · Authors · 2022-11-17
> **Feedback Incorporated in Paper Revision**
>
> Thank you again for your careful and insightful comments! We have uploaded a revised version of our paper, where we have incorporated your feedback:
> 1. We clarify our assumptions of access to workspace bounds and extrinsic rewards in Section 3.1-3.3.
> 2. We improve our notation, clarity, and wording of Section 2.2: Equivalent-Policy Invariant Comparison. Thank you for pointing out our wording; we have now corrected this.
> 3. We make various notational and typographic changes throughout the paper to ensure consistency, fluency, and clarity.

---

### Official Review · Reviewer_qUWm · 2022-10-24

**Confidence:** 4
**Correctness:** 3
**Technical Novelty And Significance:** 4
**Empirical Novelty And Significance:** 2
**Recommendation:** 6

**Clarity, Quality, Novelty And Reproducibility:**

The writing is at times hard to follow (see weaknesses above), but the introduced approach is novel and the experimental evaluation has a comprehensive set of analysis experiments.

**Strength And Weaknesses:**

# Strengths

- the problem of skill selection is important for effective fine-tuning of pre-trained agents —> the paper presents a novel approach that can work fully offline and leads to competitive results with prior works that require online interactions —> this is a nice contribution

- the use of the EPIC metric to compare rewards and *optimize* the skill choice is a novel idea and potentially useful even beyond the discussed approach, e.g. similar problems of downstream task reward alignment occur in unsupervised meta-RL approaches and aligning rewards with this robust metric could be an interesting direction to explore there too

- the submission has comprehensive analysis experiments that visualize the EPIC loss for different trajectories and give a good intuition that it correlates well with downstream task success. I also appreciate the ablation studies on other, more naive reward comparison metrics that nicely justifies the added complexity of the EPIC loss


# Weaknesses

The submission’s main weaknesses are in terms of the writing and the experimental evaluation:

(A) **writing hard to follow, too many details in appendix**: the description of the method as well as the experimental evaluation is a bit hard to follow at times. One reason is that many details are pushed to the appendix, which makes It hard to understand the paper from following the main text. For example, the very beginning of the method section should first give a rough intuition for how unsupervised skill discovery methods with intrinsic rewards work and introduce how they train a discriminator to diversify skills. Instead, the paper assumes that readers are already familiar with these concepts. Similarly, in the experimental section, it should be more explicitly stated what the standard skill selection method from prior work is — I had to go through the URLB paper (Laskin’21) to figure out that grid search over skills with the first 4k environment steps is the standard approach — that should be mentioned prominently in the submission. Also baselines, like the SeqEnv are only explained in the appendix, which makes it very hard to understand the comparisons in the experimental evaluation.

(B) **experimental results not very strong**: the presented experimental results show that the proposed method merely matches prior works and on most of the URLB tasks a random skill selection baseline is performing similarly. While prior works do require some environment interactions for skill selection, they use only 4k interaction steps, which is a small fraction of the 100k steps that are used for fine-tuning. Thus, the effective fine-tuning efficiency gains in Fig 3 are marginal.

(C) **discussion of assumptions and limitations not explicit enough**: the paper does not clearly state the implicit assumptions of their method in comparison to prior works. In contrast to prior works, the proposed method requires access to the downstream reward function (as a trade-off vs not requiring online interactions), it assumes that we can sample a representative distribution of states (this would be very hard for image-based observations for example) and for the sequential task evaluations assumes access to per-subtask reward functions. Its effectiveness seems limited in sparse reward cases and, in contrast to prior work, cannot straightforwardly be applied to image-based environments. While it is okay to have tradeoffs like this with respect to prior work, they should be more explicitly discussed to make readers aware of them when choosing what method to use.



# Questions

- How does the proposed approach perform with sparse rewards? This needs some more discussion (see point on limitations above)

- How does the method perform with shorter offline pre-training? URLB shows performance with 100k, 500k, 1M pertaining steps too — would be good to understand the tradeoffs of the proposed approach vs prior skill selection approaches in this regime too.

- Do you see a way to extend the proposed approach to environments with image-based observations?

- What are the assumptions in the sequential task case? Do you assume access to a reward function for every sub-task?



**Summary Of The Paper:**

There is many works that propose methods for unsupervised skill discovery, e.g. via behavior diversification. This work builds on top of these and proposes an approach for offline skill selection when using the unsupervisedly pre-trained agent for downstream tasks. Specifically, assuming access to a downstream task reward function, they find a skill that aligns the intrinsic reward used during unsupervised pre-training with the downstream task reward under a robust reward metric, EPIC. In experimental evaluations on the URLB benchmark they show that this skill selection is competitive with prior works that require online environment interactions.

**Summary Of The Review:**

The paper proposes a novel and interesting approach for a relevant problem. The introduced idea of using the EPIC loss to optimize for reward alignment potentially is useful even outside the proposed application, so I think the paper can be an interesting contribution for the community. I do think that the writing could be substantially improved to make it easier to follow and the experimental results don’t show a substantial improvement over prior work. Thus, in conclusion, I am supporting acceptance but it’s only a weak accept and I am open to change my mind based on the other reviewers’ comments.

---

> ### Author Response · Authors · 2022-11-10
> **Apt questions and interesting extensions of method addressed**
>
> Thank you for both the positive feedback regarding novelty and analysis as well as some insightful comments regarding writing clarity and experiments. We address the criticisms and questions below.
>
> **Readability and details in appendix**
>
> We appreciate the comments regarding refactoring some important experimental details as well as background knowledge around skill learning into the main paper for ease of reading. We will look into addressing all of the raised suggestions here prior to the conclusion of the rebuttal period. We will also look into improving the overall fluency of the writing to make the work easier to follow.
>
> **Experimental Results**
>
> Current skill pretraining algorithms do not yet consistently provide a sufficiently rich skill library to enable positive transfer on every task in URLB and thus the full potential of the gains from optimal skill selection on sample-efficiency are yet to be realized. Particularly for the Walker environment, which is the most challenging of the continuous control environments due to its instability, most of the learned skills are very far from having positive transfer to the tasks with the exception of standing. For these settings, choosing a skill optimally is just as good as random. However, as skill pretraining methods improve on these more complex tasks, the performance gaps between skills will increase as will the premium on finetuning samples further increasing the appeal of interaction-free skill selection methods. We hope that principled finetuning algorithms like IRM will encourage further improvements to unsupervised RL pretraining methods.
>
> Furthermore, IRM performs skill selection with 0 environment interactions. The episode length of the URLB environments is 1000, meaning that in order to evaluate a single skill, rollout based methods must exhaust 1000 environment steps (i.e. grid search spends 1000 * 10 = 10,000 environment steps 10% of the available finetuning budget). By contrast, our method immediately uses new environment steps for improving the policy. We will correct our omission to shift all of the rollout based methods to the right in Figure 7.
>
> **Assumptions Compared to Prior Work**
>
> While we do mention some of these tradeoffs in the description of our method, we agree that this could be done with more clarity and in direct comparison to baselines and prior work. We will look into making this discussion explicit in a revised version of the work before the rebuttal period ends.
>
> **Sparse Rewards**
>
> As mentioned in the paper, the sparse reward setting might be more challenging for matching based approaches due to the discontinuous nature of the reward function. Rollout based approaches also inherit challenges in skill selection as they are reliant on a rolled-out skill fully solving the task. For simpler tasks like Jaco this may work at times, however, in general, more flexible sampling strategies such as sampling positions uniformly at random in the workspace present a more promising methodology towards extracting signal in sparse settings. This is because we can practically scale up the number of offline samples to get greater coverage and increase the likelihood of sampling rewarded states, whereas, we are constrained to our skill policy’s state visitation in order to get sparse signal when using rollout-based methods. Moreover, we will look into providing additional analysis into the EPIC landscape for such settings before the conclusion of the rebuttal period.
>
> **Pretraining Length Experiment**
>
> While we agree that it would be interesting to assess trade-offs of various finetuning methods with different pretraining budgets, we thought it to be somewhat tangential to our central contribution. We center our claims around the setting where there has been sufficient pretraining to fully learn a rich skill library and thus train skill pretraining until convergence at 2 million steps. However, we will consider including this result if time permits.

---

> ### Author Response · Authors · 2022-11-10
> **Apt questions and interesting extensions to method addressed**
>
> **Extension to Image-Based Observations**
>
> We defer a full extension of IRM to skill-based RL from pixels to future work. However, we do recognize a promising approach forward could be to include an image generative model component at pretraining. In order to obtain realistic image samples to compute the EPIC loss, one could learn an expressive generative model such as a VAE over the image states obtained during pretraining and sample from the model to generate diverse and realistic sampled states. For state-based rewards, we might learn an image-reward model by regressing the rewards encountered during exploration as is done in the World Model literature (e.g. Dreamer). Given recent work employing image-based rewards learned offline from large-scale, passive data, this represents an exciting area of future research towards increasing the autonomy of our learning systems.
>
> **Sequential Environment Assumptions**
>
> You are correct, we do assume these things - we will be more explicit about this in a revised version of the paper we will upload before the conclusion of the rebuttal period.
>
> Overall, we look forward to addressing the many insightful suggestions as well as improving the readability of the work.

---

> ### Author Response · Authors · 2022-11-17
> **Feedback Incorporated in Paper Revision**
>
> Thank you again for your suggestions and comments! We have uploaded a revised version of our paper, where we have incorporated your feedback:
> 1. We have restructured and clarified our **Section 2.1 Unsupervised Skill Pretraining** to include the mutual information breakdown and standard skill selection protocol. We hope this provides a more comprehensive background.
> 2. We have additional information about our **sequential skill selection methods** in Section 4.3.
> 3. In Section 3.3, we clarify our assumptions for **sequential fine-tuning**.
> 4. We have emphasized **limitations and assumptions** of our method throughout the paper: Section 3.1 - 3.3, Section 4, Conclusion.
> 5. In Appendix A.12.4, we show the EPIC loss landscape for **sparse rewards**. While shaped reward is ideal for our method, the EPIC loss landscape can be smooth for sparse rewards. We note that a slightly higher tolerance for the sparse reward yields similar heat plots to our main figure.

---

### Official Review · Reviewer_FyAF · 2022-10-25

**Confidence:** 4
**Correctness:** 3
**Technical Novelty And Significance:** 4
**Empirical Novelty And Significance:** 4
**Recommendation:** 6

**Clarity, Quality, Novelty And Reproducibility:**

### Clarity, Quality
The presentation of the paper was mostly clear. Few unclear parts :
- Section 3.3 is very difficult to follow without more formal description or example.
- Figure 3 starts plotting from the middle only to denote the number of samples needed for skill selection, but have done only for Env Rollout CEM. In my understanding, Env Rollout and Grid Search also need to be started from the middle.

A few unimportant flaws are found :
- Multiple emphasizing in Table 1 "Walker Stand" row
- Inconsistent decimal format in Table 2

### Reproducibility
Code is not provided, but implementation details and hyperparameters are described.



**Strength And Weaknesses:**

### Strength
- The proposed method can be a powerful tool that can facilitate zero-shot deployment of unsupervised trained skill policy into actual target tasks.
- The idea of matching intrinsic reward and extrinsic reward seems novel and the use of EPIC loss is well-motivated. The analysis supports that EPIC is effectively matching the intrinsic reward and the target task reward.

### Weakness
- Missing baseline
    - Given access to the extrinsic reward function, one can use it to relabel pre-training time buffer data. Then, the skill that is relabeled with the highest reward can be selected. I think this relabeling-based skill selection is the most convincing apples-to-apples comparison because it doesn’t use the idea of intrinsic and extrinsic reward matching but does use the extrinsic reward function.
    - HRL method deserves to be a baseline for long-horizon skill sequencing
- Empirical support is limited.
    - Most of the comparison is not showing any consistent tendency. It is rather showing that the proposed skill selection can work worse than the grid search method or sampling-based skill selection. (table 1 all rows except Fetch Push, Figure3 first row, Figure 7).
    - Skill sequencing experiments only show results of IRM random. Considering the inconsistency over IRM optimization methods in Table 1, the result should be presented along with other IRM optimization methods, or a reasonable model selection criteria should be given, to conclude IRM is better.



**Summary Of The Paper:**

The paper proposes a method, IRM, that finds appropriate skills for a given target task by matching the unsupervised-learned intrinsic reward and the extrinsic reward of the target task.
The matching procedures don’t require environment samples as in conventional fine-tuning approaches but assume black-box access to the ground truth reward function of the target task.
EPIC loss is used to correctly measure the behavior similarity between the intrinsic reward and the extrinsic reward.
In their evaluation, IRM shows the same order of zero-shot and fine-tuning performance with the baselines.

**Summary Of The Review:**

The proposed method is very convincing and supported by analysis results.
However, provided empirical results less support the effectiveness of the core idea.
Current results are enough to show that IRM is a working idea,
but the benefit of IRM, over any naive approach leveraging extrinsic reward function, is not clearly shown.
Thus, I vote to reject this submission for now as I think the important comparison is missing,
but still willing to raise my score according to the rebuttal discussion.

### Post rebuttal
I appreciate the authors' efforts in the response.
The rebuttal response resolved my main concern about the experimental results.
I'm convinced that IRM is the only method that can zero-shot select skills without accessing pre-training data.
Still, given relabeling baseline provides a good intuition of the benefit of accessing the extrinsic reward function and how well EPIC is leveraging it.
Thus, I raised my rating as I agree with sharing the idea of EPIC-based skill matching with the community and leaving the study of consistently working skill matching methods to future work.

---

> ### Author Response · Authors · 2022-11-09
> **Some notes on several insightful suggestions**
>
> Thank you for providing several insightful suggestions regarding how we might improve our work. We are in the process of addressing many of them with some additional results. In the meantime, we wanted to reply with a few notes.
>
> **Pretraining Data Relabelling Baseline**
>
> Regarding the proposed reward relabeling baseline, we considered a similar approach, however, opted for other baselines for the following reasons:
> - Skill rollout durations during pretraining are often quite different from the full episode lengths of the downstream task making for a challenging evaluation
> - The skill policies change quite a bit throughout the course of unsupervised pretraining, and therefore much of the pretraining data will be outdated, no longer reflecting the final skill policy’s behavior
> - The baseline, like grid search, is unable to select arbitrary skills in continuous space apart from those sampled during pretraining
> - Relabelling does not represent as scalable an approach since storing pretraining data can be highly memory intensive.
>
> Nevertheless, we agree that such a baseline that leverages the extrinsic reward is a compelling comparison and thus will look into providing this result before the end of the revision period to further substantiate our contribution.
>
> **Consistent Experimental Trends and Benchmarking**
>
> Regarding the comment about consistent experimental trends we make two notes. Firstly, current skill pretraining algorithms do not yet consistently provide a sufficiently rich skill library to enable positive transfer for every task, and thus the full potential of the gains from optimal skill selection on sample-efficiency are yet to be realized. As skill pretraining methods improve, the performance gaps between skills will increase as will the premium on finetuning samples. We hope that principled finetuning algorithms like IRM will encourage further improvements to unsupervised RL pretraining methods.
>
> Furthermore, IRM performs skill selection with 0 environment interactions. The episode length of the URLB environments is 1000, meaning that in order to evaluate a single skill, rollout based methods must exhaust 1000 environment steps (i.e. grid search spends 1000 * 10 = 10,000 environment steps - 10% of the available finetuning budget). By contrast, our method immediately uses new environment steps for improving the policy. We will correct our omission to shift all of the rollout based methods to the right in Figure 7.
>
> **HRL Baseline**
>
> We agree that an HRL baseline would make for an interesting comparison. Our focus was mainly to extend IRM as an alternative to such methods relying on expensive environment interactions. We will look into providing results of an HRL baseline before the conclusion of the revision period that provide further support for IRM as an alternative framework in the long-horizon setting.
>
> Given that we were more interested in the validity of the IRM framework on sequential skill selection than optimizing performance here, we opted for a naive random search for the IRM optimization. However, we will look into providing additional experiments with other optimization methods as suggested.
>
> **Additional Comments**
>
> The multiple boldings are to account for best performers within the standard error bars as opposed to best average performance. We will provide code for reproducibility, correct typographical flaws, and provide additional clarity in section 3.3.

---

> ### Author Response · Authors · 2022-11-17
> **Feedback Incorporated in Paper Revision**
>
> Thank you again for your careful and insightful comments! We have uploaded a revised version of our paper, where we have incorporated your feedback:
> 1. We have added a **reward-relabelling baseline** in Section 4 and Table 1. We believe that there may be many flaws to this method, as discussed in our previous comment, but this still constitutes a relevant and useful baseline. We hope this additional baseline clarifies the performance of our method.
>
> | **Task**         | **IRM CEM**    | **IRM GD**      | **IRM Rand**   | **Env Roll.**    | **Env CEM**      | **GS**          | **Relabel**     | **Rand**         |
> |------------------|----------------|-----------------|----------------|------------------|------------------|-----------------|-----------------|------------------|
> | Jaco Top Left    | 0.000 +-0.00   | 0.000 +-0.00    | 0.000 +-0.00   | 0.186 +-0.11     | 0.770 +-0.28     | **1.84 +-0.00** | 0.000 +-0.00    | 0.000 +-0.00     |
> | Jaco Top Right   | 0.0860 +-0.040 | 0.640 +-0.24    | 0.120 +-0.097  | 7.34 +-3.4       | 9.82 +-5.3       | **16.1 +-0.00** | 0.000 +-0.00    | 3.50 +-2.5       |
> | Jaco Bot. Left   | 0.0520 +-0.030 | 0.000 +-0.00    | 0.000 +-0.00   | 0.175 +-0.16     | **0.408 +-0.22** | 0.102 +-0.00    | 0.000 +-0.00    | 0.000 +-0.00     |
> | Jaco Bot. Right  | 2.48 +-2.2     | 0.000 +-0.00    | 0.360 +-0.31   | 0.086 +-0.073    | **9.07 +-3.3**   | 0.191 +-0.00    | 0.000 +-0.00    | 0.00100 +-0.0010 |
> | Walker Stand | **19.9 +-9.3** | 9.75 +-1.4      | 12.5 +-3.0     | 18.9 +-3.7       | **22.4 +-4.3** | 13.9 +-4.4      | 3.00 +-0.18     | 20.8 +-7.6       |
> | Walker Walk | 5.86 +-0.34    | 7.48 +-0.55     | **15.5 +-5.5** | 14.9 +-2.9       | 13.3 +-3.2       | 9.40 +-2.8      | 4.99 +-1.3      | **15.6 +-4.9**     |
> | Walker Run       | 6.82 +-0.66    | 7.17 +-0.28     | 8.10 +-0.97    | 7.92 +-0.69      | 5.87 +-1.2       | 6.56 +-1.2      | 2.67 +-0.25     | **8.81 +-1.2**  |
> | Walker Flip      | 20.6 +-1.2     | 14.8 +-1.1      | 17.3 +-2.3     | **23.8 +-1.9** | 17.3 +-2.8       | 21.8 +-0.00     | 3.29 +-0.00     | 14.4 +-1.8       |
> | Quadr. Stand   | **51.5 +-12** | 40.3 +-11       | 40.2 +-13      | 40.6 +-9.7       | 47.5 +-8.7       | 37.4 +-12       | **56.1 +-11**  | 44.6 +-13        |
> | Quadr. Run  | 23.9 +-5.5     | **24.4 +-4.8** | 20.2 +-6.5     | 20.6 +-4.4       | 24.2 +-4.1       | 17.3 +-5.5      | **24.9 +-6.5** | 21.7 +-6.2       |
> | Quadr. Jump | 38.1 +-8.5     | **41.1 +-9.4** | 35.9 +-11      | 30.5 +-7.0       | 36.8 +-6.3       | 29.7 +-9.8      | 35.5 +-9.0      | 33.3 +-9.5       |
> | Quadr. Walk      | 17.5 +-6.2     | 11.5 +-3.7      | 17.1 +-6.2     | 19.3 +-2.7       | 25.5 +-4.0       | 9.21 +-2.0      | **31.3 +-8.2** | 16.4 +-5.8       |
> | Fetch Reach      | 95.9 +-1.0     | 87.5 +-0.20     | 92.5 +-1.1     | 85.0 +-6.2       | 87.8 +-1.9       | **97.3 +-0.00** | 43.9 +-0.00     | 16.7 +-19        |
> | Fetch Push  | **80.2 +-2.5** | 73.1 +-0.48     | 77.6 +-2.7     | 74.3 +-0.92      | 75.4 +-2.6       | 72.1 +-0.00     | 23.6 +-0.023    | 51.5 +-12.5      |
>
>
>
> 2. We have added an **HRL Baseline** in Section 4.3, Table 2, and Figure 3. We hope this serves as a useful baseline for more complex hierarchical RL methods.
> 3. We have added **Skill-Sequencing w/ Different IRM Optimizations** in Section 4.3, Table 2, and Figure 3.
>
> | **Task**        | **IRM Rand Seq** | **IRM CEM Seq** | **IRM GD Seq** | **Env Seq** | **HRL**     |
> |-----------------|------------------|-----------------|----------------|-------------|-------------|
> | Fetch Reach Seq | 88.1 +-1.5       | **89.5 +-0.34** | 86.7 +-0.64    | 80.7 +-4.7  | 28.4 +-31.0 |
> | Fetch Push Seq  | **84.9 +-0.12**  | **84.9 +-0.12** | 81.4 +-1.9     | 83.7 +-0.30 | 78.9 +-3.1  |
>
>
> 4. We have clarified **skill sequencing** and provided a concrete example in Section 3.3.
> Figure 3 includes shifts for environment-based methods, including Environment Rollout and Grid Search. However, because these methods are confined to the 4k skill selection budget (URLB), the shift is negligible on the diagram. Env Rollout CEM uses many more environment samples due to the optimization necessary.

---

### Author Response · Authors · 2022-11-17
**Paper Revision**

Thank you to all of the reviewers for providing invaluable feedback as to how we can improve both the intellectual merit and communication in our submission. We have uploaded a tentative draft, with edits including many improvements to our work that we hope you would consider in your final evaluations.
- We have restructured and clarified our **Section 2.1 Unsupervised Skill Pretraining** to include the mutual information breakdown and standard skill selection protocol. We hope this provides a more comprehensive background.
- We clarify and correct notation errors in **Section 2.3 Equivalent-Policy Invariant Comparison and Algorithm 1**. We hope this clarifies any confusion of our theoretical contributions.
- In **Section 3.2 and 3.3**, we clarify assumptions of our method: knowledge of workspace bounds and access to extrinsic rewards.
- In **Section 4.1**, we strengthen our argument by emphasizing the sample-efficiency of IRM over environment rollout methods and reference new results in **Figure 7**.
- In **Section 4 and Table 1**, we add a reward-relabelling baseline.
- In **Section 4.3**, we add additional baselines for sequential goal-reaching. First, we add a hierarchical RL baseline. Next, we ablate the IRM sequential skill selection process with additional IRM optimization methods CEM and Gradient Descent. We present new empirical findings in **Table 2** and **Figure 3**.
- In **Appendix A.12.4**, we show the EPIC loss landscape for sparse rewards.

If there are remaining questions regarding aspects of our submission, we would be happy to address them and make additional changes before the conclusion of the rebuttal period.

We have also included an anonymous zip file with our code.

---

### Decision · Program_Chairs · 2023-01-20

**Decision:**

Reject

**Justification For Why Not Higher Score:**

The results in the paper does not convincingly show that the method is effective, combined with limited novelty.

**Justification For Why Not Lower Score:**

N/A

**Metareview: Summary, Strengths And Weaknesses:**

The paper introduces the EPIC-loss as a way of choosing which pre-trained skill is optimal, given a downstream reward function. Reviewers all agreed that the proposed idea was interesting, of interest to the community and that the clarity of the paper had greatly improved throughout the discussion phase (with scores improving as a result). Authors should also be commended for the number of ablations provided.

However, most reviewers and AC agreed that despite this, empirical results were poor and did not convincingly support the claims made in the paper, both in the zero-shot domain (Table 1) and fine-tuning regime. Fig 7 in particular is problematic as it shows that zero-shot gains are neither consistent (e.g. on par with random skills on Walker {Run,Stand}) nor significant compared to the benefits of fine-tuning. Given the claimed contribution of the EPIC loss (in contrast to the EPIC reward comparison metric), I also personally expected bigger improvements compared to IRM-Rand, especially for larger dimensional skill spaces where sampling based would underperform vs optimization based approaches (Fig 6).

While we acknowledge that these results could come down to sub-optimal skill pre-training algorithms, it does not follow that results would necessarily improve with better algorithms, nor can we base our decision on this hypothetical scenario. Ultimately, the authors chose the problem setting and it is their responsibility to show that the method works well in this setting. Had the EPIC framework been first proposed in this paper, poor empirical results could perhaps come second to the idea, however as an application of EPIC to the problem of skill selection empirical results must come first. And for that reason, I am recommending rejection at this stage but am looking forward to seeing future revisions of this work.

In a personal reviewing capacity (which I stress did not impact the above decision), I also believe that this paper would benefit from the broader context of rapid adaptation in RL, as “skill selection” is only one mechanism for adapting quickly to downstream rewards. Similar experiments could have been conducted in the successor features framework which makes me think this, or the related VISR method (if we do not want to assume known features) could have been valid baselines. Could the authors not also evaluate the EPIC loss in multi-goal conditioned RL (multi-task pre-training, with known reward functions per task), as a way to hedge against underperforming skill discovery algorithms?


**Summary Of Ac-Reviewer Meeting:**

3/4 reviewers were present, with the exception of [1F21] due to scheduling conflicts.

The main takeaways were as follows:
* method is interesting, well motivated and would be of interest to the community.
* the assumptions of known reward function, and ability to sample valid states were not deal breakers as I first expected from the reviews.
* however reviewers all agreed that the paper suffers from poor empirical results.
* no reviewer was willing to increase their score, with one saying between a binary "strong accept/reject" they would lean reject. Another [jnVw] who had increased his score to 6 due to improved clarity, was also more leaning 5 then 7.
* none of the reviewers disagreed with my statement that the paper does not convincingly show that the proposed method is effective, nor with my outlined meta-review above.